# Animate3D: Animating Any 3D Model with Multi-view Video Diffusion

**Yanqin Jiang**[1,2*]  **Chaohui Yu**[3,4*]  **Chenjie Cao**[3,4]  **Fan Wang**[3,4]
**Weiming Hu**[1,2,5]  **Jin Gao**[1,2†]

[1]State Key Laboratory of Multimodal Artificial Intelligence Systems (MAIS), CASIA
[2]School of Artificial Intelligence, University of Chinese Academy of Sciences
[3]DAMO Academy, Alibaba Group  [4]Hupan Lab
[5]School of Information Science and Technology, ShanghaiTech University

`jiangyanqin2021@ia.ac.cn`
`{huakun.ych,caochenjie.ccj,fan.w}@alibaba-inc.com`
`{jin.gao,wmhu}@nlpr.ia.ac.cn`

https://animate3d.github.io/

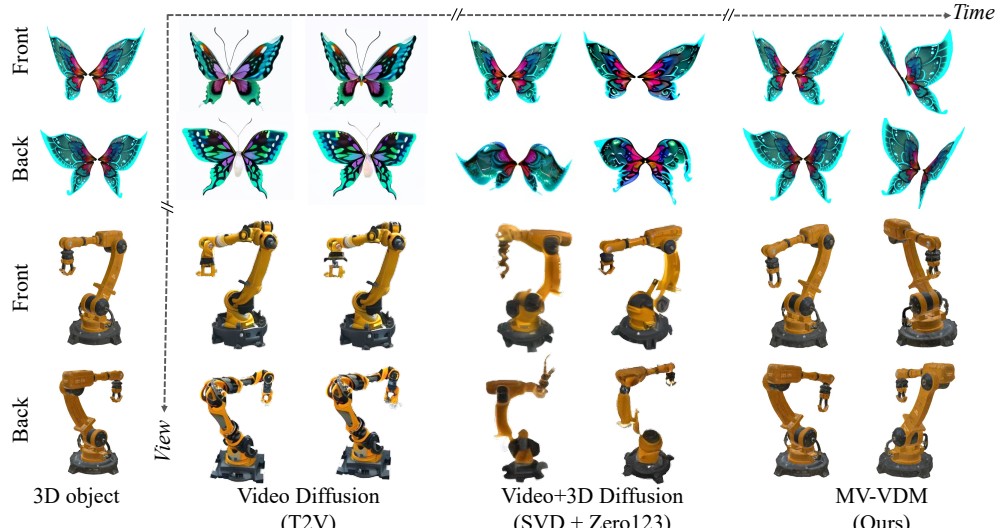

Figure 1: Different supervision for 4D generation. MV-VDM shows superior spatiotemporal consistency than previous models. Based on MV-VDM, we propose Animate3D to animate any 3D model.

## Abstract

Recent advances in 4D generation mainly focus on generating 4D content by distilling pre-trained text or single-view image-conditioned models. It is inconvenient for them to take advantage of various off-the-shelf 3D assets with multi-view attributes, and their results suffer from spatiotemporal inconsistency owing to the inherent ambiguity in the supervision signals. In this work, we present Animate3D, a novel framework for animating any static 3D model. The core idea is two-fold: 1) We propose a novel multi-view video diffusion model (MV-VDM) conditioned on multi-view renderings of the static 3D object, which is trained on our presented large-scale multi-view video dataset (MV-Video). 2) Based on MV-VDM, we introduce a framework combining reconstruction and 4D Score Distillation Sampling (4D-SDS) to leverage the multi-view video diffusion priors for animating

---

[*]Equal contribution. Work done during Yanqin's internship at DAMO Academy, Alibaba Group.
[†]Corresponding author.

38th Conference on Neural Information Processing Systems (NeurIPS 2024).

3D objects. Specifically, for MV-VDM, we design a new spatiotemporal attention module to enhance spatial and temporal consistency by integrating 3D and video diffusion models. Additionally, we leverage the static 3D model's multi-view renderings as conditions to preserve its identity. For animating 3D models, an effective two-stage pipeline is proposed: we first reconstruct motions directly from generated multi-view videos, followed by the introduced 4D-SDS to refine both appearance and motion. Benefiting from accurate motion learning, we could achieve straightforward mesh animation. Qualitative and quantitative experiments demonstrate that Animate3D significantly outperforms previous approaches. Data, code, and models are open-released.

# 1 Introduction

3D content creation has garnered significant attention due to its wide applicability in AR/VR, gaming, and movie industry. With the development of diffusion models [45, 44, 39, 20, 51, 6, 19] and large-scale 3D object datasets [13, 12, 37, 65, 70, 14], recent 3D foundational generations have seen extensive exploration through fine-tuned text-to-image (T2I) diffusion models [31, 40, 41, 52, 32, 49], as well as training large reconstruction models from scratch [22, 61, 47, 67, 54], leading the 3D assets creation to a new era. Despite the significant progress in static 3D representation, this momentum has not been paralleled in the realm of dynamic 3D content generation, also known as 4D generation.

4D generation is more challenging due to the difficulty of simultaneously maintaining spatiotemporal consistency in visual appearance and dynamic motion. In this paper, we mainly focus on two challenges: 1) *No foundational 4D generation models to unify both spatial and temporal consistency.* Though recent 4D generation works [43, 24, 5, 69, 29, 38, 63, 59, 4, 57, 66, 16] separately distill knowledge from pre-trained T2I and 3D diffusion models [39, 41, 31] and video diffusion models [51, 2, 6] to model multi-view spatial appearance and temporal motions respectively, we clarify that such a detached learning way suffers from inevitable accumulation of appearance degradation as the motion changed, as shown in Fig. 1 (SVD+Zero123). 2) *Failing to animate existing 3D assets through multi-view conditions.* With the development of 3D generations, animating existing high-quality 3D content becomes a common demand. However, previous works about 4D modeling from video [24, 57] or based on generated 3D assets [5, 29, 38, 66] are all based on text [39, 41] or *single-view* [31, 40] conditioned models, struggling to faithfully preserve multi-view attributes during the 4D generation, such as the back of butterfly in Fig. 1 is ignored by Zero123 [31].

To address these issues, we advocate for an approach better suited for 4D generation, that is, **animating any off-the-shelf 3D models with unified spatiotemporal consistent supervision**. In this way, it would be convenient to directly take advantage of various fast-developing 3D generation and reconstruction approaches based on a single foundational model, eliminating the accumulation of errors in modeling appearance and motion.

To this end, we propose a novel 4D generation framework called Animate3D in this paper, which can be divided into a foundational 4D generation model and a joint 4D Gaussian Splatting (4DGS) optimization. Formally, the foundational 4D model is a Multi-View Video Diffusion Model (MV-VDM) built upon the 3D generation model, MVDream [41], which can synchronously synthesize multi-view images with various temporal motions. Specifically, to better inherit the prior in previous 3D and video diffusion models trained on large-scale data, we propose a learnable plug-and-play spatiotemporal attention module, building upon the motion module in video diffusion [18, 6, 20] to expand the attention learning from the temporal domain to the spatial and temporal domain. Moreover, MV-VDM also includes the ability to refer to multi-view images, sufficiently preserving the identity and details of off-the-shelf 3D assets. Specifically, given multi-view images rendered from existing 3D assets or collected from real-world objects, we expand adaptive image-to-video work, I2V-Adapter [17], to multi-view version, called MV2V-Adapter, incorporating multi-view conditions to 4D learning through additionally spatial features and text embeddings. Enhanced by the multi-view appearance injection, we can disentangle the appearance learning from the motion learning, ensuring MV-VDM focuses on learning natural and coherent dynamic motions.

To further enable impressive animations from 3D objects that can be observed at any viewpoint and time, we jointly optimize the 4DGS [55] through both reconstruction and 4D Score Distillation Sampling (4D-SDS) losses based on our unified MV-VDM. Benefiting from the spatiotemporal

consistent multi-view video generations, 4DGS can be roughly converged to proper results with only reconstruction loss, while 4D-SDS further improves the details and fine-grained motions. The Gaussian trajectory learned by our framework is surprisingly accurate and could be used to directly animate the mesh.

The main dilemma in building a foundational 4D generation model lies in the rarity of large-scale 4D datasets, which is non-trivial to collect but the key factor to drive our MV-VDM. In this work, we make the first attempt to build a large-scale multi-view video (4D) dataset, dubbed MV-Video. Specifically, MV-Video comprises about **84K** animations that are available under a public license, consisting of about **38K** animated 3D objects at all, which are rendered into over **1.3M** multi-view videos with minigpt4-video [3] generated prompts, to serve as the training dataset for our 4D foundation model.

We highlight the contribution of this paper as follows: 1) Animate3D is the first 4D generation framework to animate any 3D objects with detailed multi-view conditions. The framework is further extended to achieve mesh animation without skeleton rigging. 2) We propose the foundational 4D generation model, MV-VDM, to jointly model spatiotemporal consistency. 3) We present the largest 4D datasets MV-Video collected with about 84K animations and over 1.3M multi-view videos. Extensive experiments demonstrate that our data-driven approach can generate spatiotemporal consistent 4D objects, significantly outperforming previous counterparts.

## 2   Related Work

**3D Generation.** Early 3D generation works optimized single 3D object with CLIP loss [36] or Score Distillation Sampling (SDS) [33] from 2D text-to-image (T2I) diffusion models. Since the models providing supervision lacked 3D prior, those works usually suffered from spatial inconsistency, *i.e.*, multi-face Janus problem [50, 28, 53]. To tackle this problem, on the one hand, some works [31, 40, 41, 32, 35] lifted the T2I diffusion to multi-view image diffusion by injecting new spatial attention layers and fine-tuning on large-scale 3D synthetic datasets [13, 12]. Although 3D consistency was improved, these optimization-based methods still required a relatively long time to optimize a 3D object. On the other hand, some feed-forward 3D generation foundation models [22, 61, 47, 67, 54, 60, 30], also trained on large-scale 3D datasets, were able to produce a good-quality 3D object in several seconds in an inference-only way. Inspired by the success of the data-driven approaches in 3D generation, we aim to construct a large-scale 4D generation dataset and take the pioneering step towards developing foundation models for 4D generation.

**Video Generation.** Video generation works started with text-to-video (T2V) generation [20, 42, 2, 18, 6, 19, 7, 10], subsequently followed by image-to-video (I2V) approaches [64, 17, 58, 6]. Previous T2V works usually built upon T2I diffusion models [20, 42, 19, 18, 21], leveraging their pre-trained weights by leaving the spatial blocks unchanged and inserting new temporal blocks to model the temporal camera or object motions. The I2V works [58, 17, 64], building upon the aforementioned T2V methods, typically incorporate image semantics into video models. This is achieved through cross-attention mechanisms between noisy frames and the conditional image, while retaining the motion module design from the T2V models unaltered. We draw inspiration from the development paradigm of video generation to design our 4D generation foundation model, which is a multi-view image conditioned multi-view video diffusion model building upon pre-trained multi-view 3D and video diffusion models.

**4D Generation.** The pioneering work of 4D generation is MAV3D [43], which is a text and image-conditioned 4D generation framework. MAV3D first proposed a multi-stage pipeline to optimize the static 3D object generation through the T2I model and subsequently learn motions from the T2V model [42]. Following works [68, 5, 29, 69, 4, 59] adopted a similar pipeline, and they further found that employing T2I [39] and 3D-SDS [41] are crucial for both object generation and motion learning stages. Without them, the quality of the generated object's appearance suffered a remarkable decline, and the motion-learning process was prone to failure. Very recently, Consistent4D [24] proposed a video-to-4D generation task, which used single-view video reconstruction and SDS from Zero123 [31] for motion and appearance learning. This paradigm was adopted by following works [38, 63, 66, 57, 16, 11] and extended to text/image-to-video then video-to-4D generation. All aforementioned works heavily depend on the foundational model for SDS to preserve objects' appearance and attributes. However, existing 3D diffusion models struggle to refer to multi-view

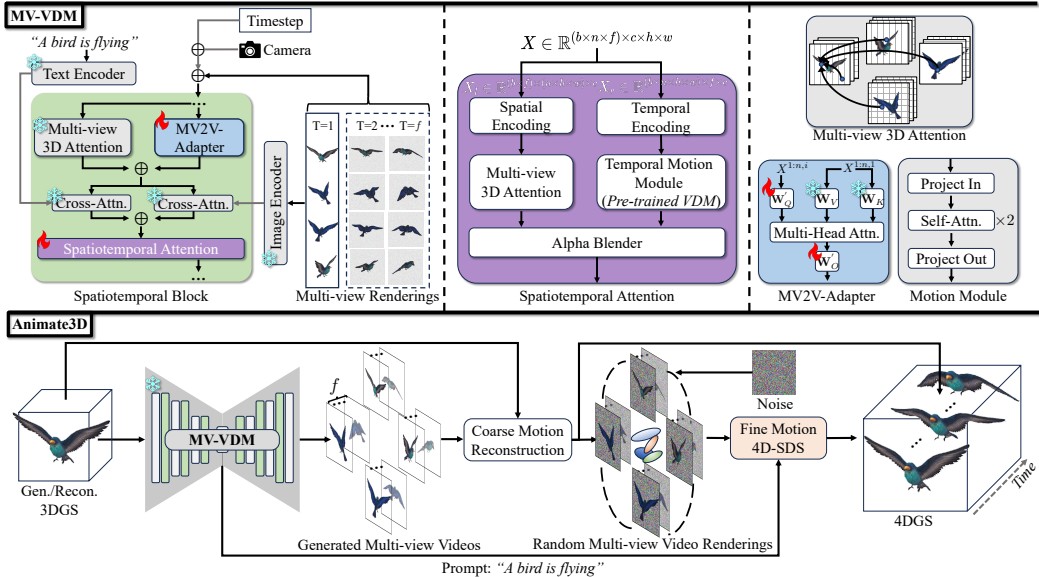

Figure 2: Illustration of our proposed multi-view video diffusion model—**MV-VDM** (upper part) and our **Animate3D** framework (lower part). MV-VDM, trained on our presented large-scale 4D dataset MV-Video, can generate spatiotemporal consistent multi-view videos. Animate3D, based on MV-VDM, combines reconstruction and 4D-SDS optimization to animate any static 3D models.

conditions, restricting their broader applications to animate various off-the-shelf 3D assets without losing their multi-view attributes.

Furthermore, it is worth noting that existing 4D generation methods suffer from another issue, *i.e.*, spatial and temporal inconsistency [43, 5, 29, 24, 38]. Because the diffusion models used for SDS were never trained with multi-view video (4D) datasets, missing the critical capacity to formulate spatial and temporal consistency simultaneously. Thus previous methods failed to properly trade off a good balance between appearance and motion learning. Please refer to Sec. B in appendix for detailed discussion and comparison.

In this work, we resort to disentangle 3D object generation/reconstruction and motion learning through a foundational 4D generation model, and propose a novel framework to animate any static 3D object with consistent multi-view attributes.

## 3 Method

Given a static 3D model, our goal is to animate it with a text prompt and use its multi-view renderings as image condition. This 4D generation task is particularly challenging as it requires ensuring spatial and temporal consistency of the appearance and motion, compatibility with the prompt, and preserving the identity of the static object. To address these challenges more fundamentally, we propose a novel framework, Animate3D, to animate any static 3D object. As depicted in Fig. 2, we divide the task into two parts: learning a multi-view video diffusion model (MV-VDM), and animating 3D object with multi-view videos generated by MV-VDM.

### 3.1 Multi-view Video Diffusion Model (MV-VDM)

We propose a novel multi-view image conditioned multi-view video diffusion model, named MV-VDM. To inherit prior knowledge acquired by spatially consistent 3D models and temporally consistent video models trained on large-scale datasets, we advocate a baseline architecture by integrating them to utilize their pre-trained weights. In this work, we take MVDream [41] and AnimateDiff [18] for the 3D and the video diffusion model, respectively. To enhance the spatiotemporal consistency and ensure compatibility with the prompts and the object's multi-view images, we propose an efficient plug-and-play spatiotemporal attention module combined with an image-conditioning approach. Our

MV-VDM is trained on our presented large-scale multi-view video dataset, MV-Video, which is introduced in Sec. 4.

**Spatiotemporal Attention Module.** As illustrated in Fig. 2, we insert a novel spatiotemporal attention module after cross-attention layers. The proposed spatiotemporal attention module comprises two parallel branches: the left branch is for spatial attention, and the right branch is for temporal attention. For spatial attention, we adopt the same architecture as the multi-view 3D attention in MVDream [41]. Specifically, the original 2D self-attention layer is converted into 3D by connecting $n$ different views. In addition, we incorporate 2D spatial encoding, specifically sinusoidal encoding, into the latent features to enhance spatial consistency. As for temporal attention, we keep all designs of the temporal motion module from the video diffusion model [18] unchanged in order to reuse their pre-trained weights. Based on the features of these two branches, we employ an alpha blender layer with a learnable weight to achieve features with enhanced spatiotemporal consistency. It is worth noting that we do not apply spatiotemporal attention across all views and frames due to the prohibitive GPU memory requirements that render training infeasible. Instead, our parallel-branch design offers an efficient and practical alternative. Specifically, we first reshape the input feature of spatiotemporal attention module $X \in \mathbb{R}^{(b \times n \times f) \times c \times h \times w}$ into two forms, $X_l \in \mathbb{R}^{(b \times f) \times (n \times h \times w) \times c}$ for spatial branch and $X_r \in \mathbb{R}^{(b \times n \times h \times w) \times f \times c}$ for temporal branch. The spatiotemporal attention is then computed as:

$$X_{out} = \mu \cdot \text{Attention}_{\text{spatial}}(X_l W_Q^s, X_l W_K^s, X_l W_V^s) W_O^s +$$
$$(1 - \mu) \cdot \text{Attention}_{\text{temporal}}(X_r W_Q^t, X_r W_K^t, X_r W_V^t) W_O^t, \tag{1}$$

where $\mu$ denotes the learnable weight, $W_Q^{s/t}, W_K^{s/t}, W_V^{s/t}, W_O^{s/t}$ represent the corresponding projection matrices. $b, n, f, h, w, c$ are the batch size, views, frames, height, width, and channels of the image features, respectively.

**Multi-view Images Conditioning.** Inspired by I2V-Adapter [17], we add a new attention layer, termed MV2V-Adapter, parallel to the existing frozen multi-view 3D self-attention layer within the proposed spatiotemporal block, as shown in Fig. 2. Concretely, noisy frames are first concatenated along the spatial dimension. These are then used to query the rich contextual information from the multi-view conditional frames, which are extracted using the frozen 3D diffusion model. Next, we add the output of the MV2V-Adapter layer to that of the original multi-view 3D attention layer of MVDream. Thus, for each frame $i \in \{1, ..., f\}$, denoting the multi-view input, output, and conditional frames' features as $X^{1:n,i}$, $X^{1:n,i}_{out}$, and $X^{1:n,1}$, we have:

$$X_{out}^{1:n,i} = \text{Attention}(X^{1:n,i} W_Q, X^{1:n,i} W_K, X^{1:n,i} W_V) W_O +$$
$$\text{Attention}(X^{1:n,i} {W_Q}', X^{1:n,1} W_K, X^{1:n,1} W_V) {W_O}', \tag{2}$$

where $W_Q, W_K, W_V$ and $W_O$ are projection matrices in original self-attention layer, while ${W_Q}'$ and ${W_O}'$ are those in the newly added layer. We find this simple cross-attention operator can effectively improve the object's appearance consistency in the generated video. After that, as shown in the spatiotemporal block in Fig. 2, we employ two cross-attention layers to align the text prompt and preserve the object's identity, respectively. The left one is inherited from MVDream, while the right one is pre-trained in IP-Adapter [62].

**Training Objectives.** The training process of our multi-view video diffusion model is similar to Latent Diffusion Model [39]. Specifically, the sampled multi-view video data $q_0^{1:n,1:f}$ are first encoded into latent feature $z_0^{1:n,1:f}$ via encoder $\mathcal{E}$ frame by frame and view by view. Then we add noise using the forward diffusion scheduler: $z_t^{1:n,2:f} = \sqrt{\bar{\alpha}_t} z_0^{1:n,2:f} + \sqrt{1 - \bar{\alpha}_t} \epsilon$, where $\alpha_t$ is a weighted parameter and $\epsilon$ is Gaussian noise. Note that, following I2V-Adapter, we keep the first frame, *i.e.*, the condition multi-view frames clean, and only add noise to the rest of the frames. During training, the proposed MV-VDM takes as input the clean latent code $z_0^{1:n,1}$, noisy latent code $z_t^{1:n,2:f}$, text prompt embedding $y$, and the camera parameters $\Sigma^{1:n}$, and outputs the noise strength, supervised by $\mathcal{L}_2$ loss. The training objective of our MV-VDM is calculated as:

$$\mathcal{L}_{\text{MV-VDM}} = \mathbb{E}_{\mathcal{E}(q_0), y, \epsilon \in \mathcal{N}(0,I), t} [||\epsilon - \epsilon_\theta(z_0^{1:n,1}, z_t^{1:n,2:f}, t, y, \Sigma^{1:n})||_2^2], \tag{3}$$

where $\theta$ denotes the diffusion model. It is important that we keep the entire multi-view 3D attention module frozen and only train the MV2V-Adapter layer and the spatiotemporal attention module to conserve GPU memory and accelerate training. Moreover, as the multi-view images of the first frame, $z_0^{1:n,1}$, serves as the condition images, we calculate the loss only for the latter $f - 1$ frames, *i.e.*, $z_0^{1:n,2:f}$.

## 3.2 Reconstruction and Distillation of 4DGS

Based on our 4D generation foundation model MV-VDM, we propose to animate any off-the-shelf 3D object. For efficiency, we take 3D Gaussian Splatting (3DGS) [25] as the static 3D object representation, and animate it by learning motion fields represented by Hex-planes, as in [55].

**4D Motion Fields.** As in 4D Gaussian Splatting (4DGS) [55], we represent the motion fields by Hex-planes [15, 8]. Denoting the static 3DGS as $\mathcal{G} = \{\mathcal{X}, \mathcal{C}, \alpha, r, s\}$, where $\mathcal{X}$, $\mathcal{C}$, $\alpha$, $r$, and $s$ represent the position, color, opacity, rotation, and scale, respectively. The motion module $\mathcal{D}$ predicts changes in position, rotation, and scale for each Gaussian point in frame $i$ by interpolating the Hex-planes $R$. The motion fields computation can be formulated as:

$$\mathcal{F} = \bigcup_l \prod_\zeta \texttt{interp}(R^\zeta, (\mathcal{X}, i)), \tag{4}$$

$$\Delta\mathcal{X} = \phi_{\mathcal{X}}(\mathcal{F}), \Delta r = \phi_r(\mathcal{F}), \Delta s = \phi_s(\mathcal{F}), \tag{5}$$

where $l$ equals to the scales in Hex-plane, and $\texttt{interp}()$ denotes interpolating the Gaussian points on the specific plane $\zeta$ to obtain corresponding motion features. We have $\zeta \in \{(x,y), (x,z), (y,z), (x,t), (y,t), (z,t)\}$. Therefore, Gaussian $\mathcal{G}'$ at time $t$ is updated as follows:

$$\mathcal{G}' = \{\mathcal{X} + \Delta\mathcal{X}, \mathcal{C}, \alpha, r + \Delta r, s + \Delta s\}. \tag{6}$$

To better preserve the appearance of static 3D objects, we keep certain attributes, specifically opacity $\alpha$ and color $\mathcal{C}$, unchanged.

**Coarse Motion Reconstruction.** Based on the spatiotemporal consistent multi-view videos generated by MV-VDM, we first leverage a 4DGS reconstruction stage to directly reconstruct the coarse motions. Specifically, we use a simple but effective $\mathcal{L}_2$ loss as our $\mathcal{L}_{\text{rec}}$, which is calculated as:

$$\mathcal{L}_{\text{rec}} = \sum_{i=1}^n \sum_{j=1}^f \|\mathcal{C} - \widehat{\mathcal{C}}\|^2, \tag{7}$$

where $\mathcal{C}$ and $\widehat{\mathcal{C}}$ denote the multi-view and multi-frame renderings and the corresponding ground truth. As verified in Fig. 3, this reconstruction stage can already learn high-quality coarse motions by leveraging the generated multi-view videos of MV-VDM.

**4D-SDS Optimization.** To better model the fine-level motions, we introduce a 4D-SDS optimization stage to distill the knowledge of our multi-view video diffusion model. The 4D-SDS loss $\mathcal{L}_{\text{4D-SDS}}$ is a variant of $\mathbf{z}_0$-reconstruction SDS loss and can be formulated as:

$$\mathcal{L}_{\text{4D-SDS}}(\mathcal{G}, \mathcal{D}, z = \mathcal{E}(g(\mathcal{D}(\mathcal{G})))) = \mathbb{E}_{t,\Sigma,\epsilon}[\|z - \hat{z}_0\|_2^2], \quad \hat{z}_0 = \frac{z_t - \sigma_t \epsilon_\theta}{\alpha_t}, \tag{8}$$

where $z$ and $z_0$ are latent feature of the rendered image and the estimation of clean latent feature from current noise prediction $\epsilon_\theta$, respectively, $g$ represents the rendering function. $\alpha_t$ and $\sigma_t$ are the signal and noise scale controlled by the noise scheduler, respectively.

**Training Objectives.** In addition to $\mathcal{L}_{\text{rec}}$ and $\mathcal{L}_{\text{4D-SDS}}$, we introduce a variant of As-Rigid-As-Possible (ARAP) loss [46] to facilitate the rigid movement learning as well as the maintenance of the high-quality appearance of the static object. The ARAP loss $\mathcal{L}_{\text{arap}}$ in our work is defined as:

$$\mathcal{L}_{\text{arap}}(p_j) = \sum_{i=2}^f \sum_{k \in \mathcal{N}_{c_i}} w_{j,k} \|(p_j^i - p_k^i) - R_j((p_j^1 - p_k^1)\|^2, \tag{9}$$

where $\hat{R}_j$ is estimated from a rigid transformation using Singular Value Decomposition (SVD) according to [46]:

$$\hat{R}_j = \text{argmin}_{R \in \mathbf{SO}(3)} \sum_{k \in \mathcal{N}_{c_i}} w_{j,k} \|(p_j^i - p_k^i) - \hat{R}_j((p_j^1 - p_k^1)\|^2. \tag{10}$$

$\mathcal{N}_{c_j}$ denotes the set of points within a fixed radius of $p_j$, and $w_{j,k} = \exp(-\frac{d_{jk}}{d})$ where $d_{jk}$ is the distance between center of $p_j$ and $p_k$, measuring the impact of $p_k$ on $p_j$. This loss encourages the generated dynamic object to be locally rigid, and it enhances the learning with rigid movement.

In summary, the training objectives for animating off-the-shelf 3DGS object is:

$$\mathcal{L} = \lambda_1 \mathcal{L}_{\text{rec}} + \lambda_2 \mathcal{L}_{\text{4D-SDS}} + \lambda_3 \mathcal{L}_{\text{arap}}, \tag{11}$$

where $\lambda_1$, $\lambda_2$, and $\lambda_3$ are weighted parameters.

## 3.3 Extension to Mesh Animation

To directly utilize high-quality mesh generated from commercial 3D generation tools and crafted by human experts, we extent our framework to mesh animation, producing animated mesh compatible with standard 3D rendering pipelines.

We initialize the 3DGS representation of the given object by vertices and triangles of the static mesh. Specifically, the color is determined by vertex color and we average the connected edges for the scale. Opacity and rotation are set to fully visible and zero rotation quaternion, respectively. The coarse 3DGS is animated following the motion reconstruction steps as described in the above sections. We utilize the per-vertex Gaussian trajectory to deform the static mesh in a straightforward way without skeleton rigging, control point selection or complicated deformation algorithms. As shown in Fig. 6 and our project page, the results are surprisingly good despite the simplicity of the solution.

# 4 Experiment

## 4.1 Setup

**Training Dataset.** To train our MV-VDM, we build a large-scale multi-view video dataset, MV-Video. Concretely, we render multi-view videos of **37,857** animated 3D models collected from Sketchfab [1]. Each model has **2.2** animations on average, resulting in **83,716** animations in total. Each animation is 2 seconds long at 24 fps. *Note that animated models that are not allowed to be used to generate AI programs are filtered.* The statistical information of our MV-Video dataset is reported in Table 1. For more details about the rendering settings and data examples, please refer to our Appendix ( D). We will release this dataset to further advance the field of 4D generative research.

Table 1: Statistical information for our multi-view video (MV-Video) dataset.

| Model ID | Animations | Avg. Animations per ID | Max Animations per ID | Multi-view Videos |
|---|---|---|---|---|
| 37,857 | 83,716 | 2.2 | 6.0 | 1,339,456 |

**Implementation Details.** We sample 8 frames evenly for each animation to train our MV-VDM. We use the Adawm optimizer with a learning rate of $4e-4$ and a weight decay $0.01$, and train the model for 20 epochs with a batch size of 2048. When inference, we set the sampling step to 25 and adopt freeinit [56] to get stable results when animating 3D objects. As for 4D generation, the resolution and feature dimension of the Hex-planes are set to $[100, 100, 8]$ and 16, respectively. We perform coarse motion reconstruction for the first 1000 iterations with a batch size of 32 (4 views, 8 frames), and then add 4D-SDS optimization for another 400 iterations. Learning rate is 0.0015 initially and decreases linearly to 0.0005 at the end of reconstruction stage. $\lambda_1$, $\lambda_2$ and $\lambda_3$ in Eq. 11 are set to 10.0, 0.01 and 0.5 , respectively. It costs 3 days to train our MV-VDM on 32 80G A800 GPUs, and the optimization for 4D generation takes around 30 minutes on a single A800 GPU per object.

**Evaluation Dataset.** For the evaluation of MV-VDM, we render multi-view images from 128 static 3D objects and then generate multi-view videos conditioned on them. Following the evaluation setting of VBench [23], we use four different random seeds for each object and report the average results. For the evaluation of 4D generation, we generate 25 objects across various categories using the large 3DGS reconstruction model GRM [61]. Input images for GRM and animation prompts used in this work are provided in our Appendix ( E.1).

**Evlaution Metrics.** We adopt the evaluation protocol proposed in VBench [23], which is a popular video generation benchmark consisting of both T2V and I2V evaluation tools. The I2V evaluation protocol contains 9 evaluation metrics, and we choose 4 for our evaluation, *i.e.*, `I2V Subject`, `Motion Smoothness`, `Dynamic Degree`, and `Aesthetic Quality`, measuring the consistency with the given image, the motion smoothness, the motion degree, and the appearance quality, respectively. We abbreviate them as **I2V**, **M. Sm.**, **Dy. Deg.** and **Aest. Q.** in the experiment section. Values of all metrics are the higher, the better, except for `Dynamic Degree`, since we observe that sometimes completely failed results present extremely high dynamic degree. For more details about the introduction and calculation of the evaluation metrics, please refer to our Appendix ( E.2).

**Comparison Methods.** We compare our work with 4Dfy [5] and DreamGaussian4D (DG4D) [38] on the task of animating any given 3D object using their official implementations. They represent the state-of-the-art in 4D generation methods, starting by generating a static 3D object using 3D-SDS in

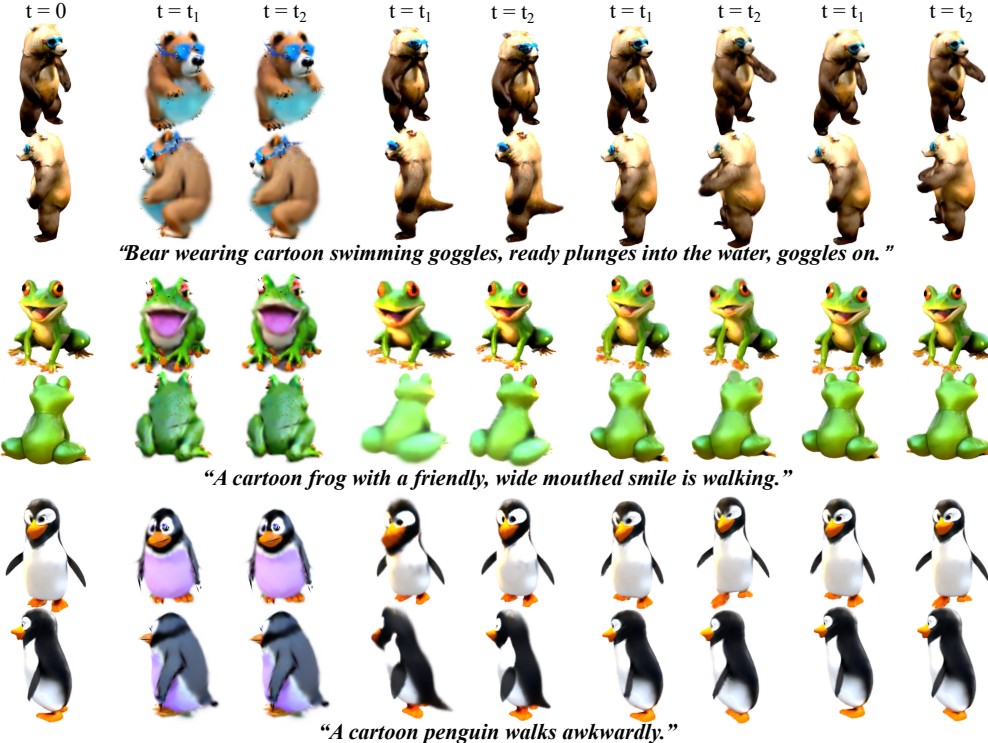

| Input | 4Dfy(Gau.) | | DG4D | | Reconstruction (Ours) | | Ours | |
|:---:|:---:|:---:|:---:|:---:|:---:|:---:|:---:|:---:|
| t = 0 | t = t₁ | t = t₂ | t = t₁ | t = t₂ | t = t₁ | t = t₂ | t = t₁ | t = t₂ |

*"Bear wearing cartoon swimming goggles, ready plunges into the water, goggles on."*

*"A cartoon frog with a friendly, wide mouthed smile is walking."*

*"A cartoon penguin walks awkwardly."*

Figure 3: Qualitative comparison with state-of-the-art methods.

the initial stage, and subsequently animating it via video SDS and single-view video reconstruction in later stages. For 4D representation, 4Dfy and DG4D adopt dynamic NeRF [34, 26] and 4DGS [55], respectively. For a fair comparison, we replace the dynamic NeRF in 4Dfy with 4DGS used in both our work and DG4D, and also apply ARAP loss for motion regularization. For DG4D, we keep the 4DGS representation and motion regularizations in their work unchanged.

## 4.2 Comparison with State-of-the-Art

In this section, we perform comprehensive comparisons with previous works, including quantitative and qualitative comparisons and user studies.

Table 2: Quantitative comparisons with state-of-the-art methods.

(a) Comparison on video generation metrics.

| | I2V ↑ | M. Sm. ↑ | Dy. Deg. | Aest. Q. ↑ |
|---|---|---|---|---|
| 4Dfy (Gau.) [5] | 0.783 | **0.996** | 0.0 | 0.497 |
| DG4D [38] | 0.898 | 0.986 | 0.477 | 0.529 |
| Ours | **0.982** | 0.991 | **0.597** | **0.581** |

(b) Comparison via user study.

| | Align. Text | Align. 3D. | Mot. | Appr. |
|---|---|---|---|---|
| 4Dfy(Gau.) [5] | 2.028 | 1.608 | 1.534 | 1.84 |
| DG4D [38] | 2.824 | 3.52 | 2.284 | 3.108 |
| Ours | **4.386** | **4.734** | **4.288** | **4.528** |

**Quantitative Comparison.** As shown in Tab. 2a, our method significantly outperforms 4Dfy and DG4D in terms of **I2V**, **Dy. Deg.**, and **Aest. Q.**. This indicates our generation results have good alignment with the given static 3D object (`I2V Subject`), dynamic motion (`Dynamic Degree`), and superior appearance (`Aesthetic Quality`). For `Motion Smoothness`, we slightly lag behind 4Dfy, since 4Dfy always generates nearly static results, as illustrated by the 0.0 `Dynamic Degree` in the first row of Tab 2a. Generally, our method is able to animate 3D object with smooth and dynamic motion, at almost no cost of sacrificing their high-quality appearance, facilitating customized and high-quality dynamic 3D object creation.

**Qualitative Comparison.** As shown in Fig. 3, it is obvious that 4Dfy's results are blurred and deviate much from the given 3D object, owing to the use of text-conditioned diffusion models to optimize the motion and appearance. Additionally, its generated objects are almost static. This is because at the beginning of the training process, the noisy rendered image sequence, *i.e.*, the input to the T2V model, has no temporal changes, which misleads the video diffusion model to generate almost static supervisions, as illustrated in Fig 1. For DG4D, its results align well with the given 3D object in front

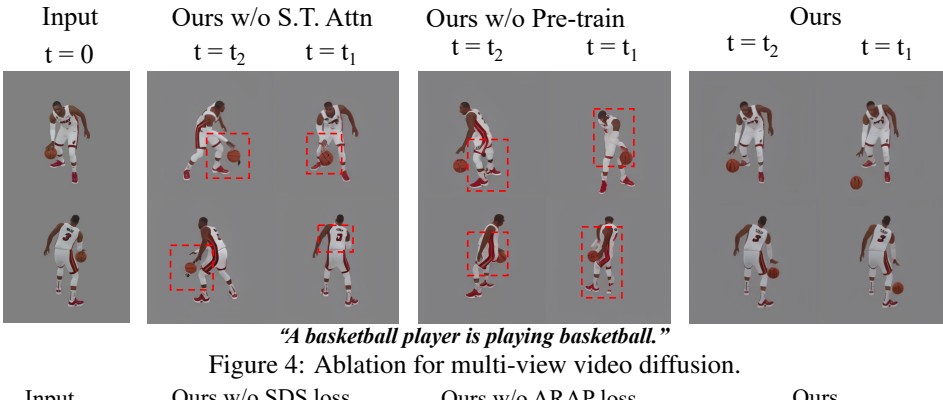

"A basketball player is playing basketball."

Figure 4: Ablation for multi-view video diffusion.

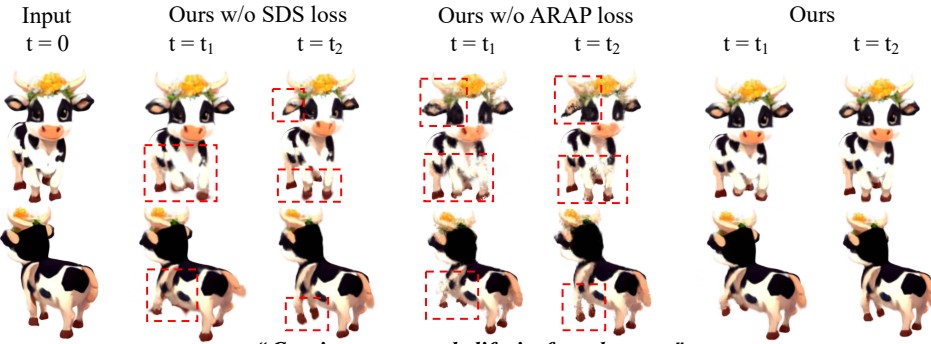

"Cow in cartoon style lifts its front hooves."

Figure 5: Ablation for 3D object animation.

view, *i.e.*, the view used for generating guided video. However, it doesn't align with the object in the novel view, as indicated by the *tail in Bear and Penguin*, *distorted goggles in bear*, and *blurred back and side views* in Fig 3. This is because it adopts Zero123 to optimize the novel views. Zero123 only conditions on the front view, leading to NVS optimization favoring pre-trained data distributions, which could lead to potential appearance degradation. More importantly, DG4D fails when the object in the guided video is assigned with movements toward the camera. For example, the frog is moving forth and back in the guided video, however, DG4D interprets it as enlargement and reduction of the object. The same thing comes to the penguin which nods towards the camera and leans forward. This misinterpretation usually results in blurry effect and strange appearance.

In contrast, our method, leveraging the spatiotemporal consistent muti-view prior, manages to deal with motion towards the camera, as demonstrated by the bear's raised front paw (our model takes the front view and its orthogonal views as the condition view, not depicted in the image). Besides, we successfully maintain the high quality appearance of the given 3D object when generating natural motion. Please refer to the videos in our *supplementary material* for a more intuitive comparison.

**User Study.** We conduct a user study among 20 people on the 25 dynamic objects and report the averaged results in Tab. 2b. The participants are asked to score the generated dynamic objects from 1 to 5, according to the alignment with the given text (**Align. Text**) and static object (**Align. 3D**), motion quality (**Mot.**), and appearance quality (**Appr.**). The user study proves the superiority of our method. Please refer to the Appendix ( E.3) for more details.

## 4.3 Ablation

Table 3: Ablation Studies

| (a) Ablation of Multi-View Diffusion | | | | | (b) Ablation of 4D Generation | | | | |
|---|---|---|---|---|---|---|---|---|---|
| | I2V ↑ | M. Sm. ↑ | Dy. Deg. | Aest. Q. ↑ | | I2V ↑ | M. Sm. ↑ | Dy. Deg. | Aest. Q. ↑ |
| w/o S.T. Att. | 0.915 | 0.980 | **0.958** | 0.531 | w/o SDS loss | 0.978 | 0.990 | **0.657** | 0.572 |
| w/o Pre-train | 0.910 | 0.981 | 0.944 | 0.531 | w/o ARAP loss | 0.970 | 0.990 | 0.573 | 0.557 |
| Ours | **0.935** | **0.988** | 0.710 | **0.532** | Ours | **0.983** | **0.997** | 0.597 | **0.581** |

**Multi-view Video Diffusion.** In Tab 3a, we validate the effectiveness of the proposed SpatioTemporal Attention (short as S.T. Att.) and the pre-trained weight from video diffusion model (short as Pre-train). We replace the proposed spatiotemporal block with temporal block from AnimateDiff, and

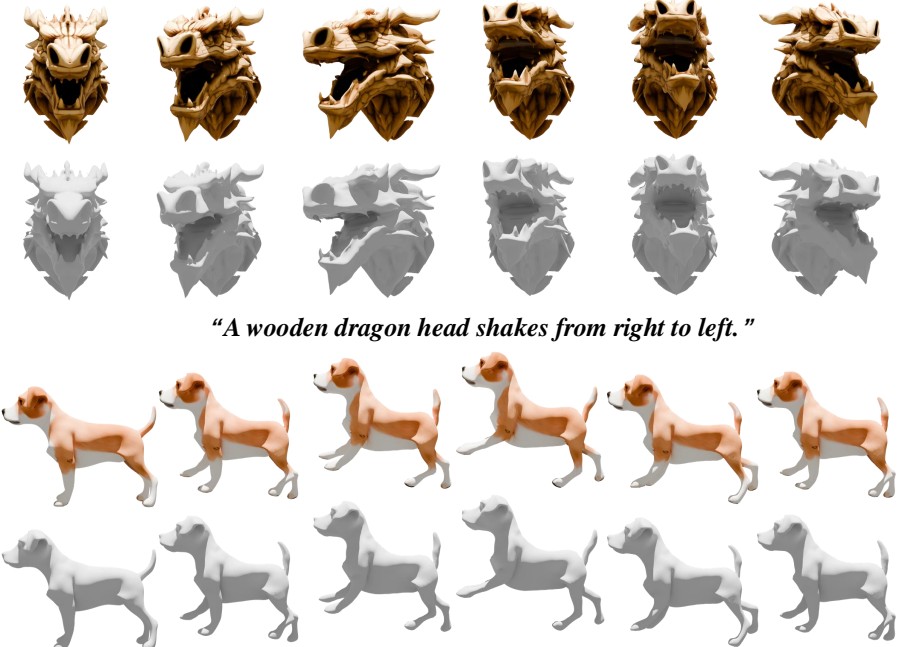

*"A wooden dragon head shakes from right to left."*

*"A cute dog is running and jumping."*

Figure 6: Visualizations of mesh animation. We present RGB and textureless renderings of two mesh animation results. Best viewed by zooming in.

this leads to performance drop in `I2V Subject`, `Motion Smoothness` and `Aesthetic Quality`. `Dynamic Degree` seems to be enhanced, but that is caused by the increase of unstable failure cases. The same tendency could be observed in experiments w/o pre-trained video diffusion weight. Therefore, we think both designs are necessary for generating multi-view videos consistent with the given multi-view images and with high-quality appearance and motion. Qualitative ablations in Fig 4 further demonstrate this point.

**4D Object Optimization.** The ablations for 4D object optimization are shown in Tab. 3b and Fig. 5. The quantitative results in Tab. 3b indicate both SDS and ARAP losses improve the alignment with the 3D object (`I2V Subject`), `Motion Smoothness`, and `Aesthetic Quality`. However, the `Dynamic Degree` decreases. We suppose the decrease in dynamic degree is mainly caused by the removal of floaters and blurry effects, which are also taken into account of dynamic degree, as shown in Fig. 5. The two losses might slightly decrease the motion amplitude, but generally, we think the overall performance is improved when applying them.

### 4.4 Mesh Animation

We provide mesh animation results in Fig. 6. Static meshes are generated by commercial 3D generation tools. For more results, please visit our project page.

## 5 Conclusion

In this work, we present Animate3D, a novel framework for animating any off-the-self 3D object. Animate3D disentangles the 4D object generation into a foundational 4D generation model, MV-VDM, and a joint 4DGS optimization pipeline based on MV-VDM. MV-VDM is the first 4D foundation model, which can generate spatiotemporal consistent multi-view videos conditioned on multi-view renderings of a static 3D object. To train MV-VDM, we present the largest multi-view video (4D) dataset, MV-Video, containing about 84K animations with over 1.3M multi-view videos. Based on MV-VDM, we propose an effective pipeline to animate any static 3D object by jointly optimizing the 4DGS via both reconstruction and 4D-SDS. Animate3D is a highly practical solution for downstream 4D applications since it can animate any generated or reconstructed 3D objects. Data, codes, and pre-trained weights will be released to facilitate the research in 4D generation.

# 6 Acknowledgements

The authors would like to thank the anonymous reviewers for their valuable comments and suggestions. This work was supported in part by the Natural Science Foundation of China (Grant No. 62192782, U22B2056, 62422317), the Beijing Natural Science Foundation (Grant No. L223003, JQ22014), the Natural Science Foundation of China (62036011, U2033210, 62102417, 62222206, 62172413), the Project of Beijing Science and technology Committee (Project No. Z231100005923046). Jin Gao was also supported in part by the Youth Innovation Promotion Association, CAS. This work was also supported by Damo Academy through Damo Academy Research Intern Program.

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

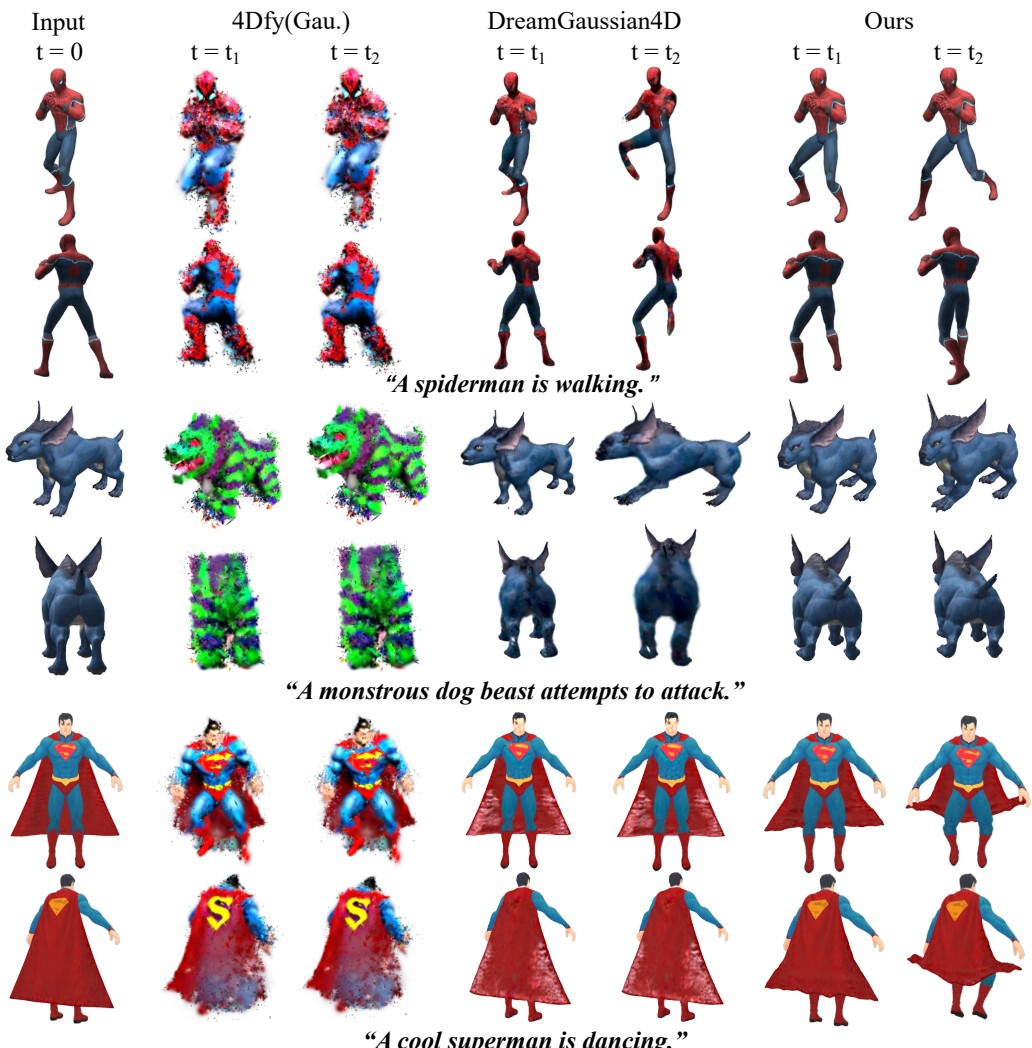

| Input | 4Dfy(Gau.) | | DreamGaussian4D | | Ours | |
| t = 0 | t = t$_1$ | t = t$_2$ | t = t$_1$ | t = t$_2$ | t = t$_1$ | t = t$_2$ |

*"A spiderman is walking."*

*"A monstrous dog beast attempts to attack."*

*"A cool superman is dancing."*

Figure 7: Qualitative comparison with state-of-the-art methods on reconstructed 3D objects.

## A   More Visualizations

We provide more qualitative results as below. Please refer to the videos attached in our supplementary materials for more intuitive visualizations.

**Comparison.** In the main paper, we only present the animation results for 3D objects generated by large reconstruction models. But our method is able to animate reconstructed objects as well as generate 3D models from text/image/video, and shows a great advantage over other methods as shown in Fig. 7. In Fig. 7, 4Dfy produces results inconsistent with the given 3D object, and they are almost static. Distortions and blurry effects could be observed in results generated by DreamGaussian4D, and it sometimes deviates from the given object in novel views. In contrast, our approach can generate results not only spatially and temporally coherent but also consistent with the input object.

**Ablation.** We provide more qualitative ablations of MV-VDM and 4D generation in Fig. 8.

## B   Discussion with respect to Previous 4D Generation Methods

Previous two-stage 4D generation works attempted to disentangle motion learning from appearance learning by adopting different types of supervision signal, i.e., video diffusion/monocular video for

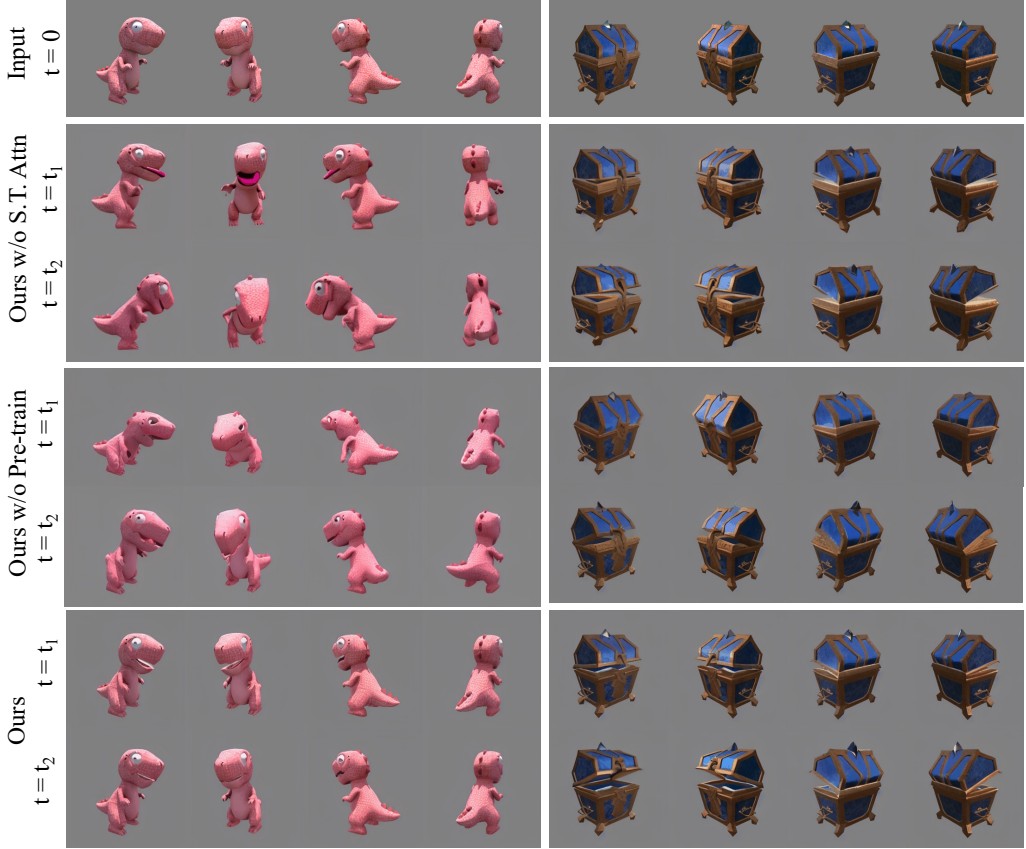

*"A pink dinosaur is waving its little hands"*      *"A blue treasure chest is being opened."*

(a) More ablation studies of MV-VDM.

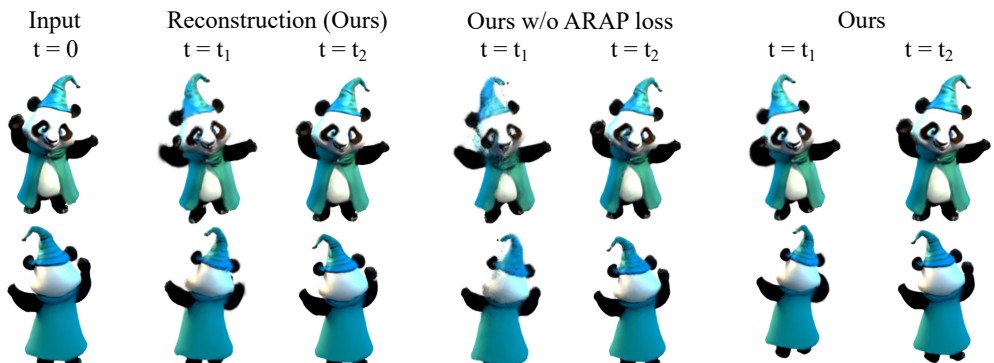

*"Magical panda in 3D cartoon style, with a wizard hat and cloak waving its hands to cast a spell."*

(b) More ablation studies of 4D generation.

Figure 8: Ablations of MV-VDM and 4D optimization.

motion and image/3D diffusion for appearance. However, the motion and appearance supervisions adopted in their work are not orthogonal, and sometimes have negative effect on each other.

For example, it is commonly agreed that video-diffusion-SDS usually brings unappealing visual effect to the appearance of the object [68, 69, 29]. Meanwhile, the appearance supervision signal prevents 4D object from updating along the direction that follows the exact score function of the video diffusion model, leading to less natural motion. Small motion amplitude in [5, 69] and shaky appearance in [29] could partly support this point. As for monocular-video-guided motion learning, previous work [38, 63, 66] rely on 3D diffusion model (Zero123 [31]) to supervise both motion and appearance in novel view. Since Zero123-SDS is applied per-frame, temporal consistency in novel view cannot be guaranteed. Moreover, monocular video doesn't provides information about depth/distance, so moving closer to or farther away from the camera can be perceived as the magnification or reduction of the object, resulting in appearance distortion.

In contrast, our method takes the unified supervision signal from MV-VDM for motion learning and appearance preservation. Our motion and appearance supervision signal inherently don't conflict with each other, since MV-VDM is conditioned on multi-view attributes of the 3D object to generate multi-view videos. Besides, multi-view motion supervision in our work enables more natural motion generation when compared with single-view motion supervision in other works. Thus, we achieve superior performance in terms of both motion generation and appearance preservation in the task of animating any off-the-shelf 3D object.

To provide a comprehensive understanding of the difference between our method and previous work, we conduct comparisons with more open-sourced previous works using their official implementations, and summarize the results in Tab. 4.

Table 4: Comparison on 4D Generation. Note that TC4D [4] takes pre-defined object trajectory as the input.

| | I2V ↑ | M. Sm. ↑ | Dy. Deg. | Aest. Q. ↑ | CLIP-I ↑ |
|---|---|---|---|---|---|
| 4Dfy [5] (4DGS) | 0.783 | **0.996** | 0.0 | 0.497 | 0.786 |
| 4Dfy [5] (NeRF) | 0.817 | 0.990 | 0.010 | 0.549 | 0.834 |
| Animate124 [68] | 0.845 | 0.986 | 0.313 | 0.563 | 0.845 |
| 4DGen [63] | 0.833 | 0.994 | 0.187 | 0.453 | 0.776 |
| TC4D [4] | 0.856 | 0.992 | **0.830** | 0.565 | 0.859 |
| Dream-in-4D [69] | 0.938 | 0.994 | 0.0 | 0.551 | 0.895 |
| DG4D [38] | 0.898 | 0.986 | 0.477 | 0.529 | 0.860 |
| Ours (8-frame) | 0.982 | 0.991 | 0.597 | **0.581** | **0.946** |
| Ours (16-frame) | **0.983** | 0.991 | 0.750 | 0.572 | 0.937 |

# C   Limitations

Despite the promising performance in generating spatiotemporal consistent 4D objects, our method still has a few limitations. First, it takes a relatively long time (about 30 minutes) to animate an existing 3D object. Second, there is a trade-off between temporal coherence and motion amplitude in the multi-view videos generated from the proposed MV-VDM. Specifically, the larger the motion amplitude, the higher the risk of temporal incoherence. Third, our model sometimes fails to animate realistic scenes due to the domain gap between synthetic training data and real-world test data. At last, current evaluation metrics in 4D generation are not sufficient, as they mainly rely on video generation metrics and user studies. Designing more suitable metrics for 4D generation will be an important future work.

# D   More Details of MV-Video Dataset

## D.1   Rendering Details.

For the rendering settings, we first centralize the model according to the bounding box of the object in the first frame. Then, we adjust the camera distance to make sure the object stays in the scope of view during the animation process. Sixteen views are evenly sampled in terms of azimuth, starting from

values randomly selected between $-11.25°$ and $11.25°$. The elevation angle is randomly sampled within the range of $0°$ to $30°$. To stabilize training, we manually filter out objects that we identify as challenging to learn due to factors such as large movements, complex environmental interactions, high speed, or sudden changes in appearance.

## D.2 Data Examples.

As shown in Fig. 12 and Fig. 13, we showcase more examples of our multi-view video dataset (MV-Video).

Furthermore, as shown in Fig. 9, we extracted all nouns from the text captions of our MV-Video dataset, which are generated by minigpt4-video [3], and plotted a word cloud of the Top-1000 nouns. We can see that our MV-Video dataset contains diverse categories of animated 3D objects, including humans, characters, animals, plants, mechanical structures, and *ect.*.

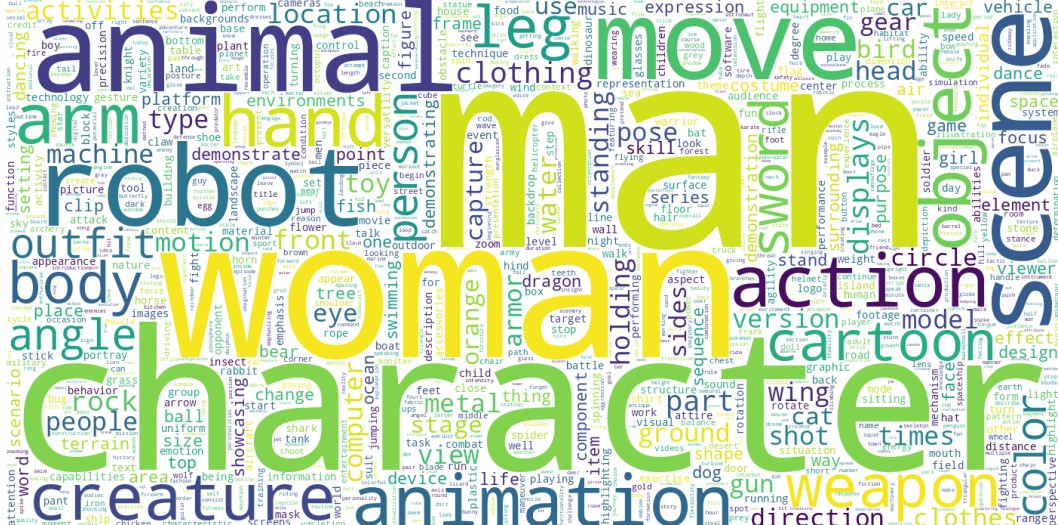

Figure 9: Illustration of the word cloud of the top 1000 nouns extracted from the text captions of our MV-Video dataset.

# E 4D Generation Evaluation

## E.1 Evaluation Dataset

In Fig. 10, we provide the input images for image-to-3D generation and corresponding text prompts for 4D animation used in Sec. 4.2.

## E.2 Evaluation Metrics

VBench [23] provides six evaluation metrics for I2V evaluation[3], *i.e.*, `I2V Subject`, `I2V Background`, `Camera Motion`, `Subject Consistency`, `Background Consistency`, `Motion Smoothness`, `Dynamic Degree`, `Aesthetic Quality` and `Imaging Quality`. Since our generated results have no background, and the evaluation cameras are fixed, metrics related to background and camera motion are not used. The metric `Imaging Quality`, which is affected by ambient light, is also not used here. `Subject Consistency` is also omitted since its calculation process is similar to `I2V Subject`, except for the choice of reference frame. It takes the first generated frame, instead of the input image used in `I2V Subject`, as the reference frame for similarity score calculation, which is not suitable for our task of animating 3D objects. We briefly introduce the evaluation metrics used in our work as below:

---

[3] https://github.com/Vchitect/VBench/tree/master/vbench2_beta_i2v

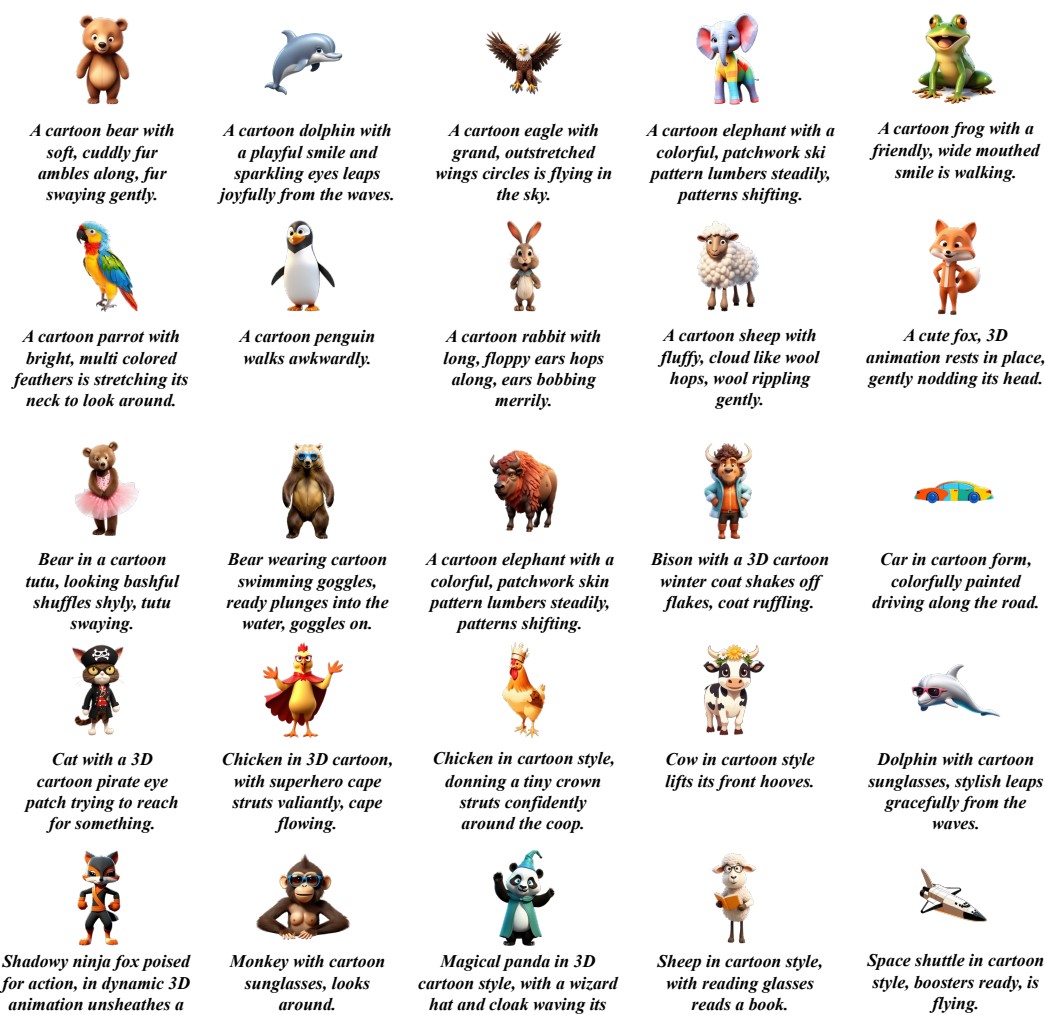

*A cartoon bear with soft, cuddly fur ambles along, fur swaying gently.*

*A cartoon dolphin with a playful smile and sparkling eyes leaps joyfully from the waves.*

*A cartoon eagle with grand, outstretched wings circles is flying in the sky.*

*A cartoon elephant with a colorful, patchwork ski pattern lumbers steadily, patterns shifting.*

*A cartoon frog with a friendly, wide mouthed smile is walking.*

*A cartoon parrot with bright, multi colored feathers is stretching its neck to look around.*

*A cartoon penguin walks awkwardly.*

*A cartoon rabbit with long, floppy ears hops along, ears bobbing merrily.*

*A cartoon sheep with fluffy, cloud like wool hops, wool rippling gently.*

*A cute fox, 3D animation rests in place, gently nodding its head.*

*Bear in a cartoon tutu, looking bashful shuffles shyly, tutu swaying.*

*Bear wearing cartoon swimming goggles, ready plunges into the water, goggles on.*

*A cartoon elephant with a colorful, patchwork skin pattern lumbers steadily, patterns shifting.*

*Bison with a 3D cartoon winter coat shakes off flakes, coat ruffling.*

*Car in cartoon form, colorfully painted driving along the road.*

*Cat with a 3D cartoon pirate eye patch trying to reach for something.*

*Chicken in 3D cartoon, with superhero cape struts valiantly, cape flowing.*

*Chicken in cartoon style, donning a tiny crown struts confidently around the coop.*

*Cow in cartoon style lifts its front hooves.*

*Dolphin with cartoon sunglasses, stylish leaps gracefully from the waves.*

*Shadowy ninja fox poised for action, in dynamic 3D animation unsheathes a glinting katana.*

*Monkey with cartoon sunglasses, looks around.*

*Magical panda in 3D cartoon style, with a wizard hat and cloak waving its hands to cast a spell.*

*Sheep in cartoon style, with reading glasses reads a book.*

*Space shuttle in cartoon style, boosters ready, is flying.*

Figure 10: Illustration of the input images for image-to-3D generation and corresponding prompts for 4D animation.

`I2V Subject` assess whether the appearance of the object in the generated video remains consistent with that in the input image. To this end, DINO [9] feature similarity across frames is calculated.

`Motion Smoothness` evaluates whether the motion in the generated video is smooth, and follows the physical law of the real world. The motion prior in the video frame interpolation model [27] is utilized for evaluation.

`Dynamic Degree` employs RAFT [48] to estimate the degree of dynamics in synthesized videos.

`Aesthetic Quality` is calculated by the LAION aesthetic predictor, which reflects the artistic and beauty value perceived by humans towards each frame.

### E.3 User Study Template

As illustrated in Fig. 11, a picture of the user study page is depicted. The survey contains 25 dynamic objects, which are shown in Fig. 10. The participants are asked to score the generated 4D objects from 1 to 5, according to the alignment with the given static object and text prompt, appearance quality, and motion quality.

**Every three constitute a group, representing the animation generations for the same object u sing three different methods (the order has been shuffled). Please compare the generated res ults within the same group and then provide scores.**

1. Please rate the animated results for the static 3D model in conjunction with the video. (The first row fea tures reference images of the 3D model, while the second row displays the generated animation results. The first and second columns represent two different viewpoints, respectively.)

   The desired text description for the animation is: "**Sheep in cartoon style, with reading glasses, reads a book.**"

   Gif

2. ...

Figure 11: The layout of our user study.

# F Border Impacts

This paper exploited 4D generation based on our proposed multi-view video diffusion model, which can generate spatiotemporal consistent multi-view videos. Because of the advanced generative capacity, our models may output misinformation or fake videos. Thus, we sincerely remind the users to pay attention to it. Note that our method only focuses on the technical aspect. All the code, dataset, and trained models will be released.

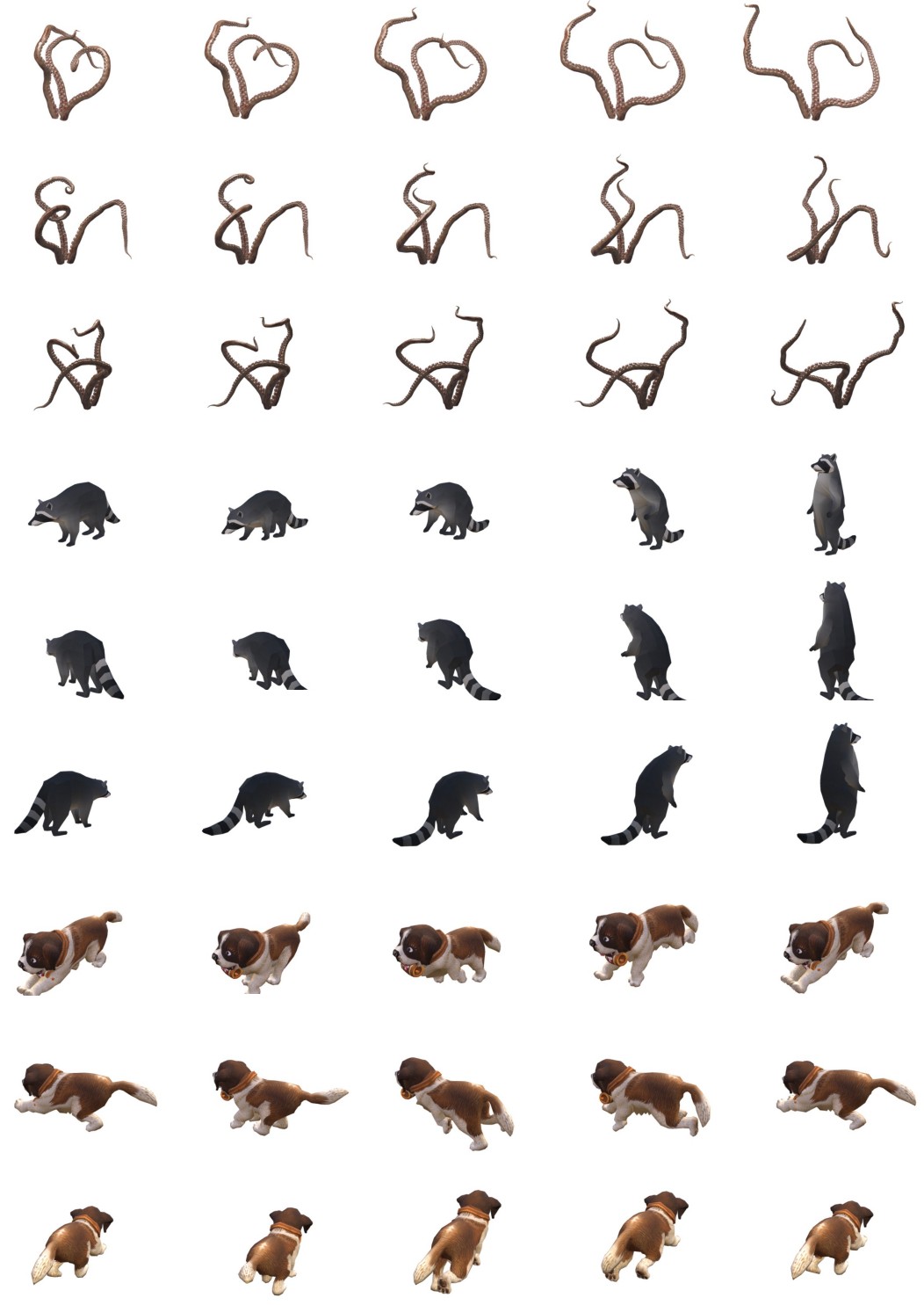

Figure 12: More examples of our MV-Video dataset.

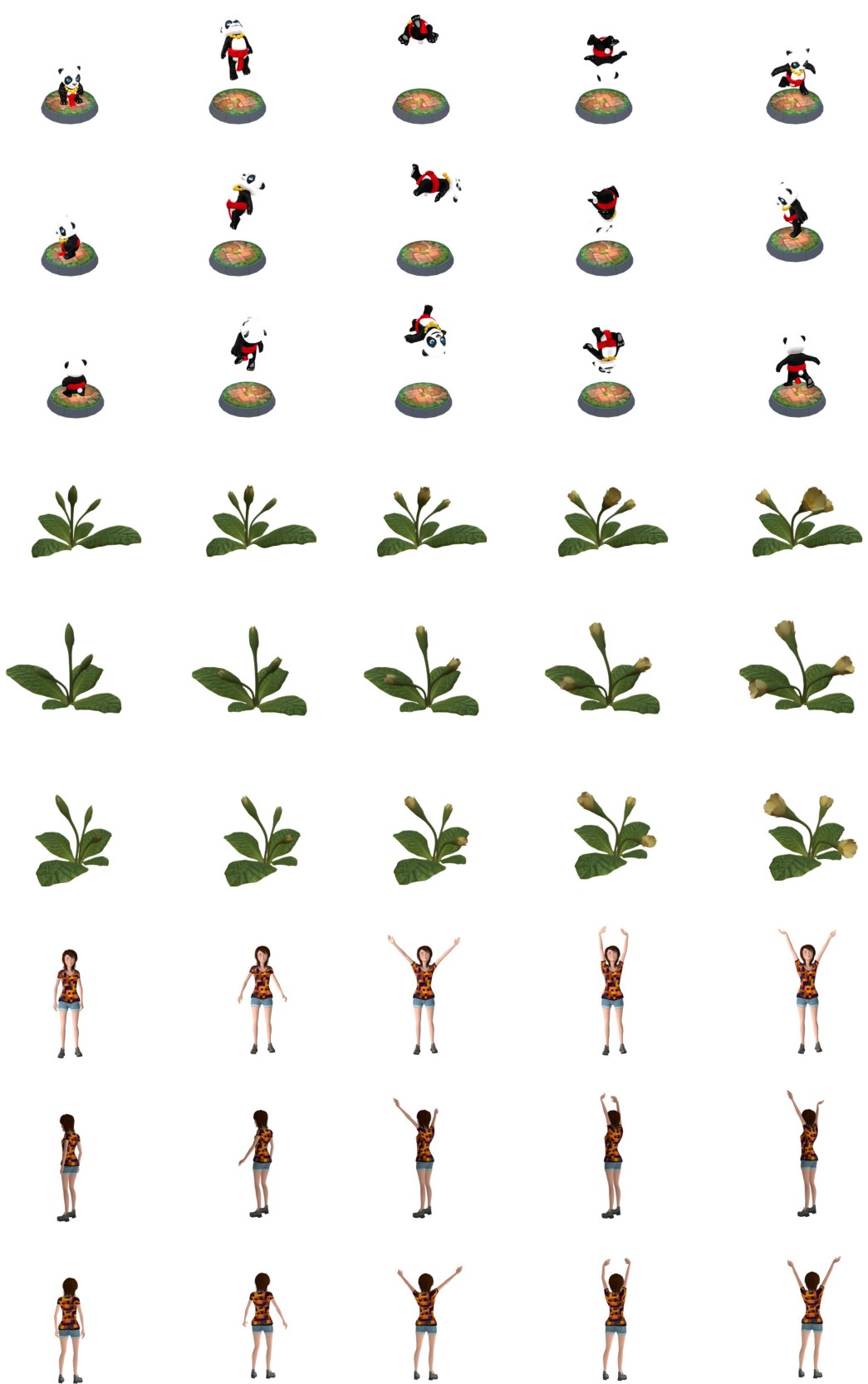

Figure 13: More examples of our MV-Video dataset.

