# OpenReview forum: "Animate3D: Animating Any 3D Model with Multi-view Video Diffusion"
_NeurIPS.cc/2024/Conference — NeurIPS 2024 poster_

### Official Review · Reviewer_2Yu3 · 2024-07-10

**Soundness:** 2
**Presentation:** 3
**Contribution:** 3
**Rating:** 6
**Confidence:** 4

**Summary:**

This paper introduces a framework for 4D generation to animate static 3D models, consisting of two components: MV-VDM, a multi-view video generation model, and a framework combining reconstruction and 4D score distillation sampling (4D-SDS). A spatiotemporal attention module enhances consistency, using multi-view renderings to preserve the model's identity. The proposed two-stage pipeline for animating 3D models first reconstructs coarse motions and then refines them with 4D-SDS. Experiments show that Animate3D outperforms previous methods.

**Strengths:**

1. The paper introduces a large-scale dataset for multi-view video generation, consisting of 84,000 animations and over 1.3 million multi-view videos, which addresses the challenge of limited 4D datasets.
2. Qualitative experiments demonstrate the ability to generate high-quality 4D objects. The generated shapes appear to have better quality and texture compared to other methods.
3. The paper is clearly written, with a clear motivation for using a multi-view diffusion model to enhance spatiotemporal consistency.

**Weaknesses:**

1. The method appears to be a straightforward engineering pipeline, combining several existing techniques (e.g., MVDream and IP-Adapter in the MV-VDM stage, reconstruction + SDS in the animating stage) without significant innovation.
2. Motion diversity is a concern. The method shifts the core problem to the quality of the multi-view video. The dataset achieves this effect, but the maximum number of animations per ID is only six, which is insufficient for capturing the diverse motions of a single 3D object. This approach lacks generality.
3. Motion controllability is limited. The results shown in the paper exhibit simple, repetitive motions with small amplitudes. Although text prompts are used as conditions, they are relatively simple, and the generated motions are not complex. The method lacks a clear approach to achieving more controllable motions.
4. Experimental results are lacking. DreamGaussian4D provides CLIP-I as a quantitative comparison to assess the consistency between prompts and generated results, which is missing in the paper. Additionally, the qualitative results provided in terms of views and moments are too few.

**Questions:**

1. Regarding time, the paper reports an optimization time of 30 minutes per object on a single A800 GPU. How much time does it take to reconstruct coarse motions and refine motions using the 4D-SDS model, respectively?

**Limitations:**

The authors have discussed the limitations and broader impacts of the study in the paper.

---

> ### Author Rebuttal · Authors · 2024-08-07
>
> **W1: Pipeline without significant innovation.**
>
> Thanks. Our work enjoys good novelty in both task formulation and solution pipeline.
> Firstly, we redefine the concept of 4D generation by introducing a novel task: animating any off-the-shelf 3D objects. This innovative task holds significant relevance in fields with limited exploration (Lines 30-41).
> Given the rapid advancements in 3D generation, our proposed task is both practical and meaningful to directly drive these high-quality 3D assets.
> Although some SDS-based methods have been investigated in previous 4D works, the community is still in need of a fast, feed-forward model specifically designed to animate 3D objects.
> Furthermore, such a model has the potential to facilitate mesh generation with PBR materials, which could enhance commercial applications in software platforms like Rodin-Gen1 and Meshy. This positions our work as a crucial step forward in the evolution of 4D generation technology.
>
> As for the novelty of pipeline, the main goal of the proposed pipeline is to **learn accurate motions of static Gaussians**. Note that unlike previous text-video-4D works, e.g., DG4D, 4DGen, STAG4D, we don't update any gaussian points and don't include gaussian densification/pruning. This is because **only in this way, we could learn the accurate trajectory of each points in static gaussian and when initializing static gaussian by the vertices of the mesh, we can achieve mesh animation, as shown in PDF attached (Figure 6).** Once we could animate mesh, 4D generation could benefit from high-quality 3D assets generated by commercial tools.
>
> **W2: Concern about the motion diversity.**
>
> Please refer to global response (3. Motion Diversity and Dataset).
>
> **W3: Concern about motion controllability.**
>
> Admittedly, text prompt could not achieve precise control. We'll include more controls such as monocular video in the future.
>
> **W4: Lack of experimental result CLIP-I; provide more views and moments for qualitative comparison.**
>
> Please refer to global response (1. Comparison methods) for CLIP-I metric. Please refer to the attached **PDF** for more visualization results.
>
> **Q1: Time cost of reconstruction and 4D-SDS.**
>
> For 8-frame version model, the Reconstruction and 4D-SDS both costs around 15 minutes, totaling 30 minutes (Line 242). For 16-frame, they both takes around 20 minutes.

---

> > ### Comment · Reviewer_2Yu3 · 2024-08-13
> > **Post Rebuttal**
> >
> > Thank the authors for addressing my questions. I am still advocating for accepting the paper.

---

### Official Review · Reviewer_G4mL · 2024-07-12

**Soundness:** 3
**Presentation:** 2
**Contribution:** 3
**Rating:** 5
**Confidence:** 4

**Summary:**

This paper proposes Animate3D, a 4D generation framework that consists of a multi-view video diffusion model (MV-VDM) followed by 4D Gaussian Splatting optimization, as well as a dataset of 38K animated 3D objects (MV-Video) that is used to provide spatiotemporal supervision to train the model. Different from text-to-4D approaches, Animate3D specifically tackles the problem of animating 3D assets which requires spatiotemporal consistency as well as identity preservation. The proposed architecture is based on a multi-view diffusion model (MVDream) and a video diffusion model (AnimateDiff) with additional spatio-temporal attention blocks. To animate a given 3D asset, the trained model is used for both reconstruction and distillation. First, a multi-view video is generated conditioned on multiple views of the asset. The video is used to optimize a 4D Gaussian Splatting resulting in a model with coarse motion. The same MV-VDM model is used for 4D Score Distillation Sampling to refine the motion. The method compares favorably to state-of-the-art 4D generation methods.

**Strengths:**

**[S1]** The problem setting in this paper is interesting and timely, in an area that has recently attracted significant attention. The paper takes a slightly different angle compared to text-to-3D, text-to-4D or video-to-4D methods, focusing on animating 3D assets based on a textual description instead of generating them from scratch.

**[S2]** This problem setting also opens up a new challenge, which is to preserve multi-view attributes during the animation. It is nice to see that the identity preservation and the animation are significantly improved compared to state-of-the-art methods.

**[S3]** The authors tackle this problem by training a multi-view video diffusion model. Given the success of image (2D), video (2D+time), and multi-view (3D) diffusion models, MV-VDM (4D) is a reasonable next step to ensure spatial and temporal consistency simultaneously.  Both the model and the dataset used to train it are likely to influence future work and could thus have significant impact.

**[S4]** The proposed architecture aims to reuse existing multi-view and video diffusion models, which are already pre-trained on a larger scale of data, thus taking advantage of the priors built into these models.

---

**Weaknesses:**

**[W1]** The data used in this paper are animated 3D models collected from Sketchfab (cited). The authors promise to distribute the collected dataset. However, there is not sufficient information provided about copyrights and licensing. The only piece of information provided is that models with the following clause are excluded:

>NoAI: This model may not be used in datasets for, in the development of, or as inputs to generative AI programs.

Based on the paper checklist, the paper is flagged for ethics review.

---

**[W2]** Lack of clarity in the method section and method figure

**(a)** The method section is very hard to parse. The notation is overloaded but not very well explained. The section would benefit from thorough revision to improve the overall clarity and quality.

**(b)** It is almost easier to understand the method through Fig. 2, but even that is dense and inconsistent, and often the notation does not align with what is written in Sec. 3.1. For example, in L175 latent features $z$ seem to be the output of the image encoder. In Fig. 2 (middle), $z$ appears to be the input to the spatiotemporal attention module. In Eq. 1, $X_l$ and $X_r$ are instead the inputs to the same module. At the same time, in Fig. 2 (left) the input to the spatiotemporal module seems to be the sum of the output of the cross-attention layers, not directly $z$.

This makes reading and understanding the method extremely difficult to the point where the reviewer cannot fully judge the technical contribution and similarities or differences to related models.

---

**[W3]** Discussion with respect to 4D generation methods could be expanded

Existing work in text-to-4D generation often splits the problem into two stages: a static 3D asset generation and a deformation field to model motion. This disentanglement makes it possible to learn the 3D motion from video diffusion models, without the need for multi-view video. The paper briefly touches upon this and compares to two 4D generation methods, 4Dfy and DreamGaussian4D. However, the paper could further elaborate on the similarities and differences of the proposed approach compared to existing methods and the advantages of a multi-view video diffusion model.

---

**[W4]** Certain architectural components are not ablated and their contribution to the overall performance is unclear. See Questions.

---

**[W5]** The temporal extent of MV-VDM is only 8 frames, which seems rather limited.

---

**[W6]** In some of the qualitative examples, the faithfulness to the text prompt is questionable. Some of the animations provided in the supplementary video are not accompanied by a text prompt, while quite a few animations appear very similar.

---

**[Minor]** Typos and suggestions:

- L103: pioneer → pioneering
- L113: All these manners aforementioned → All aforementioned work
- L131-132: this sentence can be improved for clarity since the two parts sound a bit repetitive
- L135: spatial → spatially; temporal → temporally
- L156: It should be better explained how what $X_l$ and $X_r$ represent and how these features are obtained.
- L292: levering → leveraging; manage → manages

---

**Questions:**

**Q1. [Based on W1]** Please elaborate on the difference between _3D animation via multi-view video diffusion_ and _two-stage text-to-4D approaches_. In the latter case, the second stage could be viewed as 3D animation. I understand that a multi-view video foundation model would offer spatiotemporal consistency that other approaches can likely not reach, but I would appreciate a more in-depth discussion about the difference to and the challenges of text-to-3D-and-3D-to-4D approaches than what is currently provided in the paper.

---

**Q2. [Based on W1]** Ideally, the authors should provide also empirical comparisons to additional methods (e.g., Dream-in-4D), if possible.

---

**Q3. [Based on W4]** The authors state that
>We find this simple cross-attention operator [MV2V-Adapter] can effectively improve the object’s appearance consistency in the generated video.

The effect of MV2V-Adapter should be better demonstrated with an ablation.

---

**Q4. [Based on W4]** Are the cross-attention layers described in L170-172 necessary for identity preservation and alignment with the text prompt? Their effect should be also demonstrated in the ablation studies.

---

**Q5.** The proposed 3D animation approach uses both reconstruction and distillation. It first uses the generated multi-view video to optimize a 4D Gaussian Splatting with coarse motion. Then SDS is used to refine the motion. It seems counter-intuitive that SDS would be good at modeling the finer details of the animation, since SDS is typically known for its over-smoothing behavior. The authors should further discuss the motivation behind using SDS during 4D optimization and provide additional examples to prove its effectiveness.

---

**Q6. [Based on W6]** Could the authors elaborate on the apparent lack of diversity in the generated motions? It may be helpful to provide some statistics from motion descriptions included in the dataset. Or is this a limitation of the model instead?

---

**Limitations:**

Limitations are adequately discussed in Appendix B.

---

> ### Author Rebuttal · Authors · 2024-08-07
>
> **W1: Data copyrights and licensing**
>
> We confirm that all models downloaded from Sketchfab have a distributable Creative Commons license and were obtained using Sketchfab’s public API. Besides, models marked as ``NoAI'' and restricted due to objectionable or adult thematic content were excluded from the dataset. We provide a detailed elaboration of the licensing information as below.
>
> | License               |   Number       |
> |-----------------------|---------------|
> | CC Attribution        |   37,081      |
> | CC Attribution-NonCommercial    | 460    |
> | CC Attribution-NonCommercial-NoDerivs   | 112   |
> | CC Attribution-ShareAlike       |    73 |
> | CC Attribution-NonCommercial-ShareAlike     |    66  |
> | Free Standard         |    38       |
> | CC Attribution-NoDerivs     |    18  |
> | CC0 Public Domain      |    9      |
> | **Total**                   |  37,857 |
>
> **W2: Clarity of the details of our method.**
>
> Thanks very much for careful reading. Sorry for the confusing presentation. We will revise the figures and the text in the updated version.
>
> **W3: Discussion with respect to 4D generation methods**
>
> Benefiting from the unified spatiotemporal consistent supervision of our 4D foundation model (MV-VDM), our approach can leverage the multi-view attributes of 3D objects to achieve more consistent 4D generation. In contrast, existing methods based on separated text or single-view diffusion models and video diffusion models struggle to preserve spatiotemporal consistency.
>
> More importantly, they also fail to faithfully maintain the multi-view attributes of an existed 3D object.
>
> **Due to limited space，we would like to discuss this further in the discussion phase**.
>
> **W4: More components ablations.**
>
> We provide the ablation for MV2V-Adapter, alpha blender and cross-attn layer (image) as follows. Note that we don't ablate the cross-attn layer (text) since it is a necessary component of MVDream and we freeze MVDream in our pipeline. The results indicate that all components are effective.
>
> | Method | **I2V $\uparrow$** | **M. Sm. $\uparrow$** | **Dy. Deg.** | **Aest. Q. $\uparrow$** |
> | --- | --- | --- | --- | --- |
> | w/o cross-attn layer (image) | 0.887 | 0.978 | 0.966 | 0.504 |
> | w/o alpha-blender | 0.911 | 0.982 | 0.958 | 0.528 |
> | w/o MV2V-adapter | 0.927 | 0.986 | 0.961 | 0.526 |
> | **Ours** | **0.935** | **0.988** | 0.710 | **0.532** |
>
>
> **W5: The outputs are only 8 frames**
>
> Thanks for this good point. To validate the scalability of our MV-VDM, we have trained a 16-frame version and reported the quantitative and qualitative results in global response (1. Comparison methods) and Figure 8 in the attached **PDF**, respectively. We find that 16-frame model generates larger amplitude motion while maintaining similar identity preservation ability as the 8-frame version.
>
> **W6: Faithfulness to the text prompt, similar animations**
>
> For the concerning about faithfulness of prompts, we clarify that our MV-VDM is capable of producing generations well-aligned with the prompts.
> First, our MV-Video dataset is designed to encompass a wide variety of motion prompts, as illustrated in the word cloud presented in the rebuttal **PDF** (Figure 4).
> Second, for the same object, our model can synthesize various motions according to different prompts as verified in the rebuttal **PDF** (Figure 7).
>
> However, we acknowledge that achieving extremely precise motion control through detailed prompts presents certain challenges.
> This limitation is primarily due to the constraints of the CLIP model, which serves as the text encoder for MV-VDM. It would be interesting future work to further develop more powerful 4D models for superior text alignment.
>
> Please also refer to our global response (3. Motion Diversity and Dataset) for more discussions about motion diversity.
>
> Actually, we have omitted some text prompts to make the video more clear. Most objects in the supplementary video are from our test set and the prompts are listed in Appendix (Figure 9). Text prompts for reconstructed objects  presented in the video are listed in Appendix (Figure 6). Below we list the rest text prompts:
>
> * A glowing blue butterfly is flying.
> * A cute cartoon dog is dancing.
> * A monster dog is walking.
> * A cartoon bear wearing a swimming goggles is getting ready to dive.
> * A cartoon frog is jumpping up and down.
> * A panda with very adorable features is dancing.
> * A cartoon sea lion, adorned with an expressive and charmingly animated face, is singing.
> * A eagle in cartoon style is flapping its wings.
> * A gaint panda in cartoon is walking.
> * A cool spiderman is dancing.
> * A cute lemur is dancing.
>
> **W7: Issues about typos.**
>
> Thanks, we appreciate your careful review. We will correct it in the revised paper and check if there remains any typos.
>
> **Q1: In-depth discussion between 3D animation via multi-view video diffusion and two-stage text-to-4D approaches**
>
> Please refer to W3.
>
> **Q2: empirical comparisons to additional methods (e.g., Dream-in-4D)**
>
> Please refer to global response (1. Comparison methods)
>
> **Q3: Ablation of MV2V-Adapter**
>
> Please refer to W4
>
> **Q4: Ablation: Cross-attention layers for image-condition and text prompt**
>
> Please refer to W4.
>
> **Q5: The use of 4D-SDS for fine-level animation is counter-intuitive**
>
> The role of 4D-SDS is to alleviate small floaters, similar to smoothing but without the blurry effect. Our coarse reconstruction is \textbf{sparse-view} reconstruction, i.e., we only have 4 views, and the reconstruction results are inevitably not perfect in novel views, especially when the number of Gaussian points is large. Similar SDS techniques are adopted in sparse-view 3D reconstruction works, such as ReconFusion[CVPR2023] and Sparse3D[CVPR2023]. We also provide detailed qualitative comparisons in the **PDF** attached (Figure 2).
>
> **Q6: Issues about the diversity in the generated motions.**
>
> Please refer to the global response (3. Motion Diversity and Dataset).

---

> > ### Author Response · Authors · 2024-08-13
> > **In-depth Discussion with respect to Previous 4D generation Methods**
> >
> > Due to limited space, our initial response did not permit a comprehensive discussion of previous 4D generation works, now we detail the in-depth discussion as suggested:
> >
> > Previous two-stage 4D generation works attempted to **disentangle motion learning** from appearance learning by adopting **different types of supervision signal**, i.e., video diffusion/monocular video for motion and image/3D diffusion for appearance. However, the motion and appearance supervisions adopted in their work are **not orthogonal**, and sometimes have **negative effect** on each other.
> >
> > For example, it is commonly agreed that video-diffusion-SDS usually brings unappealing visual effect to the appearance of the object [Animate124, Dream-in-4D, AYG]. Meanwhile, the appearance supervision signal prevents 4D object from updating along the direction that follows the exact score function of the video diffusion model, leading to less natural motion. Small motion amplitude in [4Dfy, Dream-in-4D] and shaky appearance in [AYG] could partly support this point. As for monocular-video-guided motion learning, previous work [DG4D, 4DGen, STAG4D] relies on 3D diffusion model (Zero123) to supervise both motion and appearance in novel view. Since Zero123-SDS is applied per-frame, temporal consistency in novel view cannot be guaranteed. Moreover, monocular video doesn't provides information about depth/distance, so moving closer to or farther away from the camera can be perceived as the magnification or reduction of the object, resulting in appearance distortion.
> >
> > In contrast, our method takes the **unified** supervision signal from MV-VDM for motion learning and appearance preservation. Our motion and appearance supervision signal inherently don't conflict with each other, since MV-VDM is conditioned on multi-view attributes of the 3D object to generate multi-view videos.
> > Besides, **multi-view motion supervision** in our work enables more natural motion generation when compared with single-view motion supervision in other works. Thus, we achieve superior performance in terms of both motion generation and appearance preservation in the task of animating any off-the-shelf 3D object.
> >
> > Thanks for your insightful suggestion of in-depth discussion, and we'll add this in revision. If you have any further questions, discussions are welcomed.

---

> ### Comment · Reviewer_G4mL · 2024-08-13
> **Response to rebuttal**
>
> Thank you for the detailed rebuttal. Most of my concerns have been addressed and I intend to keep a positive rating. I appreciate the extended discussion regarding existing methods and I would suggest integrating it into the main paper or the appendix.
>
> I also appreciate the authors' efforts in providing additional experiments and comparisons during the rebuttal phase.
>
> One concern that remains is still about the writing and presentation, which I hope the authors will improve substantially in their revision.

---

> > ### Author Response · Authors · 2024-08-14
> >
> > Thanks for the response. We are carefully revising our manuscript, including figures, tables, text and demo video. The revision will feature significant updates.

---

### Official Review · Reviewer_C8Es · 2024-07-18

**Soundness:** 4
**Presentation:** 3
**Contribution:** 4
**Rating:** 6
**Confidence:** 4

**Summary:**

This work presents Animate3D, a framework for animating static 3D models. The core idea involves two main components:
1. A multi-view video diffusion model (MV-VDM) conditioned on multi-view renderings of the static 3D object, trained on a large-scale multi-view video dataset (MV-Video).
2. A framework that combines reconstruction and 4D Score Distillation Sampling (4D-SDS) to utilize multi-view video diffusion priors for 3D object animation.
The animation process involves a two-stage pipeline: coarse motion reconstruction from generated multi-view videos, followed by 4D-SDS to model fine-level motions. Quantitative and qualitative evaluations show enhancements to previous methods.

**Strengths:**

1. Performance: The proposed method achieves state-of-the-art results. The experiments well validate the effectiveness of the proposed methods.

2. Clarity: The paper is well-written and easy to follow.

3. Technical Novelty: The main contributions of this paper are threefold: 1) The first 4D generation framework to animate any 3D objects with detailed multi-view conditions, which are incorporated by the proposed MV2V-Adapter.  2) The authors propose to animate any off-the-shelf 3D models with unified spatiotemporal consistent supervision, which can get better results. 3) The collected 4D dataset, MV-Video.

**Weaknesses:**

1. Missing Reference [a]: It is understandable that you did not compare Animate3D with Animate124 in this paper, as 4D-fy and DreamGaussian4D both demonstrate better performance compared to Animate124. However, it is unusual that you did not discuss Animate124 at all, given its relevance in previous comparisons by 4D-fy and DreamGaussian4D.

2. Limited Comparison Scope: The authors only compare Animate3D with 4D-fy and DreamGaussian4D, which seems insufficient. It would be more comprehensive to include comparisons with 4DGen and TC4D, as both claim superior performance over 4D-fy and have released their code. Additionally, while the modification of 4D-fy to a Gaussian representation for fair comparison is understandable, the original 4D-fy results should also be included for a thorough ablation study.

[a] Animate124: Animating One Image to 4D Dynamic Scene, \
[b] 4DGen: Grounded 4D Content Generation with Spatial-temporal Consistency, \
[c] TC4D: Trajectory-Conditioned Text-to-4D Generation.

**Questions:**

1. The results of 4D-fy appear strange. When I check the 4D-fy results in TC4D and 4DGen, they look more reasonable.

2. I think the authors should also conduct ablation studies on MV2V-Adapter.

**Limitations:**

The authors have discussed the limitations and potential negative societal impact.

---

> ### Author Rebuttal · Authors · 2024-08-07
>
> **W1: Missing Reference: Animate124**
>
> Sorry for missing Animate124, which is a great pioneering work. Comparison is in the global response (1. Comparison methods), required by other reviewers. Reference will be added in our revised paper.
>
>
> **W2: More comparisons with 4DGen, TC4D and original 4Dfy.**
>
> Thanks for your suggestion. Please refer to the global response(1. Comparison methods)
>
> **Q1: 4Dfy results is strange**
>
> Thanks for this point. Actually, 4Dfy is not designed for animating off-the-shelf 3D objects, instead, it is designed for text-to-4D (the task in TC4D/4Gen). So it doesn't fit our task and presents not so good results. Technically, we follow the default training setting in 4Dfy's official implementation for animating off-the-shelf static Gaussian/NeRF. Notably, that implementation reduces the learning rate of static Gaussian/NeRF to a relatively low value in the dynamic stage in hope that high-quality appearance would be preserved. Admittedly, when the static object is generated by image/3D SDS loss and **the exact image/3D SDS loss** is continually used in the dynamic stage, the high-quality appearance could be preserved, as shown in previous works. However, the image/3D SDS loss doesn't match the off-the-shelf 3D object in our task, so using it in the dynamic stage as the original paper results in appearance changes. As the learning rate for static NeRF is very low, the appearance change doesn't finish at the end of the training, so the results are strange.  We tried to use a higher learning rate, but found that resulted in very low **I2V** values, as the object appearance was changed completely.
>
> Besides, we found 3DGS sometimes fail to generate good results when supervised by MVDream SDS loss. This problem was discussed by other researchers in issue section of threestudio-3dgs repo.
>
> **Q2: Ablation of MV2V-Adapter**
>
> Please refer to our global response (2. Ablation of MV2V-Adapter)

---

> > ### Comment · Reviewer_C8Es · 2024-08-08
> >
> > Thanks for the efforts of the authors. They have conducted additional experiments to support their claims, and my concerns have been resolved. I will maintain my scores.

---

### Official Review · Reviewer_9MgC · 2024-07-21

**Soundness:** 3
**Presentation:** 2
**Contribution:** 3
**Rating:** 5
**Confidence:** 5

**Summary:**

This paper proposes an animation method that animates a 3D model in a 4D one. A  Multi-View image conditioned multi-view Video Diffusion Model  (MV-VDM)  is presented to generate multi-view videos from multi-view renderings of a static 3D object. The MV-VDM is leveraged to train the 4D Gaussian Splatting (4DGS), where As-Rigid-As-Possible (ARAP) loss, reconstruction loss and SDS are used as objectives. In addition, a multi-view video dataset is constructed to train MV-VDM. The paper conducts experiments to show the effectiveness of the proposed method.

**Strengths:**

1. The idea is well-motivated and straightforward.
2. A large-scale multi-view video dataset is presented.
3. A multi-view video diffusion model is presented, where spatiotemporal attention is introduced to animate multi-view images with motions.

**Weaknesses:**

1. The comparison is not very fair. The paper focuses on a new task, i.e., 3D animation, while 4Dfy focuses on  text-to-4D, which is a different task. Furthermore, text-to-4D is more challenging than 3D animation since multi-view images are unavailable.  The better performance gain of the proposed method may partly attributed to additional multi-view images. In other words, the worse performance of 4Dfy and DG4D may not be due to the methods themselves. In addition, the paper replaces dynamic NeRF in 4Dfy with 4DGS. However, some hyperparameters of 4Dfy are set according to dynamic NeRF, rather than 4DGS. Instead, the paper can compare the proposed method with AYG, which directly trains 4DGS. In addition, the paper can compare Animate234, which is an image-to-4D method.

2. The motivation for the proposed spatiotemporal attention block is not clear. Besides the temporal motion module and temporal encoding, the block introduces a multi-view 3D attention module. What is the multi-view 3D attention module used for? Why are temporal motion modules and temporal encoding not enough to generate motions for multi-view videos?

3. What is the influence of Alpha Blender on the performance of the proposed method?


4. Does the proposed method train all modules in the spatiotemporal attention block? It's not clear whether the "Multi-view 3D Attention" and "Temporal Motion Module (Pre-trained VDM)" are trained or frozen.

5. The implementation details of Alpha Blender are not clear. Is it implemented as a layer of MLP?

6. The temporal encoding is unclear. According to Figure 2, there is a temporal encoding module in the spatiotemporal attention block, but there is no description of what kind of temporal encoding is used.

7. How many views does each animated 3D object contain in the dataset? Only four orthogonal views or more?

8. Although the dataset contains 38K animated 3D objects, it is still much smaller compared to 2D video datasets. Can the proposed method trained on this dataset of this magnitude animate any 3D model?

9. Training a multi-view video diffusion model requires camera parameters. Are the camera parameters in the multi-view video dataset processed to ensure that they are consistent with the prior knowledge learned in the MVDream

**Questions:**

Please refer to my question above.

**Limitations:**

The authors have provided  limitations and  societal impact of their work

---

> ### Author Rebuttal · Authors · 2024-08-07
>
> **W1: (1) Issues about unfair comparison: 4Dfy and DG4D do not leverage multi-view images; (2) 4Dfy is based on nerf instead of 4DGS; (3) Add comparison with AYG and Animate124.**
>
> **(1)**: We proposed a new task of animating any off-the-shelf 3D object, and there is **no previous work specially designed for this task to compare with**, so we could only chose two representative 4D generation works to serve as the comparison baselines. Due to the lack of unified spatiotemporal consistent supervision, existing works (e.g., 4Dfy, DG4D) can only focus on using text or single-view image conditioned diffusion models for motion modeling. Experiments have shown that this often leads to spatiotemporally inconsistent 4D generation results. Our Animate3D is the first 4D generation framework to animate any 3D objects with detailed multi-view conditions. We believe this workflow is more suitable for generating spatiotemporal consistent 4D objects by leveraging advanced static 3D object reconstruction and generation literature.
>
> **(2)** and **(3)** Thanks for the suggestion. Please refer to our global response (1. Comparison methods)
>
> **W2: Motivation and effectiveness of the proposed spatiotemporal attention block.**
>
> To enhance the spatial and temporal consistency of our MV-VDM, we design a new spatiotemporal attention module, which mainly contains a temporal attention branch and a spatial attention branch. This block is build upon the motion module of video diffusion model to inherit the temporal prior. In the early experiments, we found that simply adopting the temporal attention branch is not enough. This is because the features are likely to lose multi-view consistency after being processed by the temporal branch. Therefore, we add a parallel spatial attention branch to address this issue. The effectiveness is validated in **Tab. 3(a) and Fig. 4 in our original paper**. Note that **w/o S.T. Attn** means we replace the proposed spatiotemporal block with motion modules in video diffusion models. We will add this details in our revision.
>
> **W3: The influence of the Alpha Blender.**
>
> In our spatiotemporal attention block, we employ an alpha blender layer with a learnable weight to fuse the features from both temporal and spatial branches. In the following table, we show the influence of the alpha blender layer and the results validate that it can enhance the spatiotemporal consistency of the 4D generation results.
>
> | Method | **I2V $\uparrow$**  | **M. Sm. $\uparrow$** |  **Dy. Deg.**  | **Aest. Q. $\uparrow$**|
> | --- | ---| ---| ---| ---|
> | w/o Alpha Blender | 0.911 | 0.982 | 0.958 | 0.528 |
> | Ours | **0.935** | **0.988** | 0.710 | **0.532** |
>
> **W4: Implementation details: trainable modules in spatiotemporal attention block.**
>
> Yes, we train all modules in our proposed spatiotemporal attention block, as illustrated in Figure 2 in our paper. Thanks for your reminder, we will add these details in implementation section in our revised version.
>
> **W5: Implementation details: Alpha blender**
>
> As illustrated in Eq. (1) in our paper, we perform alpha blender with a learnable weight $\mu$, which is implemented via nn.Parameter. We will add this detail in the revised version.
>
> **W6: Implementation details: Temporal encoding**
>
> Thanks for your reminder, we adopt sinusoid positional encoding in AnimateDiff as temporal encoding. We will add these details in our revised version.
>
> **W7: Implementation details: number of views of each aniamted 3D object in our dataset.**
>
> Please refer to the Appendix (C.1 Rendering Details), 16 views are evenly sampled in terms of azimuth, starting from values randomly selected between $-11.25^{\circ}$ and $11.25^{\circ}$. The elevation angle is randomly sampled within the range of $0^{\circ}$ to $30^{\circ}$.
>
>
> **W8: Training dataset size is limited, is it capable of animating any 3D object?**
>
> Adimittedly, our dataset is much smaller than 2D video datasets, but since we inherit the prior knowledge learned by 3D and video diffusion models, our model can animate most dynamic 3D objects commonly seen in daily life. We have tested various categories of objects, including humans, mammals, reptiles, birds, insects, vehicles, weapons, etc, and obtained good results. We provide more examples in the **PDF** attached. Please also refer to our global response (3. Motion Diversity and Dataset) to see our discussion about the dataset.
>
> **W9: Implementation details: camera parameters consistent with MVDream?**
>
> Yes, the camera parameters in our multi-view video dataset are processed to ensure that they are consistent with the prior knowledge learned in the MVDream. Please refer to the Appendix (C.1 Rendering Details) in our paper, 16 views are \textbf{evenly} sampled in terms of azimuth. Specifically, during training, we randomly sample four orthogonal views to form a multi-view video for each animation, which is consistent with MVDream's camera setting.

---

> > ### Comment · Reviewer_9MgC · 2024-08-11
> >
> > Thank the authors for addressing my questions. I plan to keep my positive score.

---

### Official Review · Reviewer_Tnsn · 2024-07-22

**Soundness:** 2
**Presentation:** 2
**Contribution:** 2
**Rating:** 5
**Confidence:** 3

**Summary:**

This paper focuses on animating 3D objects with multi-view diffusion models. To improve spatial and temporal consistency, this work builds a large-scale multi-view video dataset, MV-Video and designs an attention module to enhance spatial and temporal consistency by integrating 3D and video diffusion models. To enhance animations from 3D objects, this work jointly optimizes the 4DGS through reconstruction and 4D-SDS. Experiments show the proposed method can more consistent 4D generation.

**Strengths:**

* This work builds a large-scale multi-view video dataset to facilitate the training of multi-view video diffusion models.
* The method designs an attention module to encourage spatial and temporal consistency for multi-view diffusion models.
* The approach introduces 4D-SDS to leverage multi-view video diffusion models for animating 3D objects.
* The paper is easy to read and understand.

**Weaknesses:**

* This work focuses on 3D objects, however, the paper title claims "Animating Any 3D Models". It is a bit inappropriate.
* The effectiveness of the proposed dataset is not fully validated. Can this dataset improve single-view diffusion models or multi-view image-to-3D generation models, like MVDream?
* The diversity of the proposed method is unclear. Can one object perform different actions? For example, in Figure 3, can the frog perform swimming?
* The picture quality is a bit low, it is hard for readers to distinguish the advantages of the generated objects from the proposed method.
* For animating from 3D object, this work first leverages 4DGS to reconstruct coarse motions and then uses 4D-SDS for refinement. If the method uses better 4DGS algorithms, maybe the improvement from 4D-SDS will become little.

**Questions:**

* It is better to validate the effectiveness of the proposed dataset, can it also improve single-view diffusion models (e.g., T2V) or multi-view image-to-3D generation models (e.g., MVDream). Then, can these models further improve other methods, like DreamGaussian4D?
* Although the work verify the effectiveness of the S.T attention. The effect of MV2V-adapter is not validated.
* If the approach leverages better 4DGS reconstruction methods, such as [1] for the coarse motion reconstruction, can the 4D-SDS still improve the coarse results?
[1] Yang, Zeyu, et al. "Real-time photorealistic dynamic scene representation and rendering with 4d gaussian splatting." arXiv preprint arXiv:2310.10642 (2023).
* As this work designs a multi-view video diffusion model, can the authors compare it with existing single-view video diffusion models?
* For Figure 4, both the ablation models show that the player touches the basketball. However, the full model does not show this interaction. I wonder if the full model can do better for this interaction.

**Limitations:**

The paper has discussed the limitations.

---

> ### Author Rebuttal · Authors · 2024-08-07
>
> **W1: Inappropriate title: 3D objects instead of 3D models**
>
> Thanks for the advice, we will consider revising it in the revised version.
>
> **W2: Effectiveness of the proposed dataset in video or 3D diffusion models**
>
> Given limited time, we only finetune SVD on a subset (20\%) of our dataset, and glad to see some improvements. For evaluation, we evaluate the finetuned model on the validation dataset used for the ablation of MV-VDM.
>
> | Method         | **I2V $\uparrow$**   | **M. Sm. $\uparrow$** | **Dy. Deg.** | **Aest. Q. $\uparrow$**   |
> |----------------|-----------|------------|---------------|----------------|
> | SVD            | 0.927     | 0.985      | 0.849         | 0.538          |
> | SVD-finetuned  | **0.950** | **0.990**   | 0.708         | **0.557**      |
>
>
> We also use the generated video from finetuned model as the input for DG4D in 4D generation and report the results as below.
>
> |    |   **I2V $\uparrow$**   |   **M. Sm. $\uparrow$**   |  **Dy. Deg.**  |  **Aest. Q. $\uparrow$**   |
> |---|--------|-----------|------------|------------|
> | **DG4D w/o finetuned SVD** | 0.898 | 0.986 | 0.477 | 0.529 |
> | **DG4D w/ finetuned SVD** | **0.907** | **0.989** | 0.407 | **0.533** |
>
> Please note that the decrease in **Dy. Deg.** is because SVD w/o fintuning leads to artifacts and noise in the generated input video, hence leading to the failure of 4D generation. We believe that further improvements could be achieved by fintuning it on the full training dataset.
>
> **W3: Motion Diversity: object performing different actions.**
>
> Thanks for your insightful question. Please refer to our global response (3. Motion Diversity and Dataset).
>
> **W4: About the picture quality.**
>
> Actually, the picture quality is mainly affected by the quality of static 3D generated/reconstructed object, since our method is good at preserving the appearance of the given object. We should clarify that the reconstruction quality is not the primary contribution of this paper, which is potentially influenced by the hyper-parameter adjustment of GRM (or other reconsturction method). Besides, The compression of images in PDF can lead to a certain degree of loss in image quality.
>
> We provide high-quality images in Figure 6-8 in PDF attached. Although they do not have the same quality as the original renderings at 1024$\times$1024 resolution, they are better than those provided before.
>
> **W5: Better 4DGS algorithms will make 4D-SDS unnecessary**
>
> Though MV-VDM generates spatiotemporally coherent multi-view videos as the ground truth for reconstruction, **the ground truth has only 4 views**. Existing 4DGS algorithms **cannot straightforwardly address this sparse multi-view video reconstruction.**
>
> Following your suggestion, we have experimented with an improved 4DGS reconstruction algorithm [2]. The results are reported in the table below. Better 4DGS[2] doesn't improve the reconstruction results primarily because the task here involves sparse multi-view video reconstruction using only 4 views. Better 4DGS[2] is not designed to tackle such scenarios.
> In our work, we apply arap loss to 4DGS[1] to effectively handle sparse views. We fail to apply arap loss to Better 4DGS[2] given that arap loss is designed to constrain the motion of the 3D Gaussians, however, Better 4DGS[2] doesn't learn the motion (It regards the time dimension as a property of Gaussian points, similar to scale/rotation/opacity). We find the proposed 4D-SDS improves the results of Better 4DGS. We will add this discussion in our revised paper.
>
> | Reconstruction | 4D-SDS | **I2V $\uparrow$**  | **M. Sm. $\uparrow$** |  **Dy. Deg.**  | **Aest. Q. $\uparrow$**|
> |---|---|---|---|---|---|
> | 4DGS [1] |   | 0.978 | 0.990 | **0.657** | 0.572 |
> | Better 4DGS [2] |   | 0.972 | 0.990 | 0.621 | 0.561 |
> | 4DGS [1] | $\checkmark$ | 0.983 | **0.997** | 0.597 | **0.581** |
> | Better 4DGS [2] | $\checkmark$ | **0.984** | **0.997** | 0.610 | 0.573 |
>
> [1] Wu, Guanjun, et al. "4d gaussian splatting for real-time dynamic scene rendering." (CVPR2024)
>
> [2] Yang, Zeyu, et al. "Real-time photorealistic dynamic scene representation and rendering with 4d gaussian splatting." (ICLR2024)
>
> **Q1: Validate the effectiveness of the proposed dataset.**
>
> Please refer to the response of W2.
>
> **Q2: Ablation of MV2V-adapter**
>
> Please refer to our global response (2. Ablation of MV2V-Adapter)
>
> **Q3: The effectiveness of better 4DGS algorithms.**
>
> Please refer to the response of W5.
>
> **Q4: Comparison with existing single-view video diffusion models**
>
> We conducted some empirical comparisons between our MV-VDM and SVD, finding that our model outperforms SVD when animating 3D object renderings, as we specifically trained our model in this domain. However, we struggled to animate realistic images with complicated backgrounds, which is an area where SVD excels. It’s important to note that the comparison is made between the multi-view video results of our model and the single-view video results of SVD, which may be an unfair assessment for our model. For quantitative evaluation, we tested SVD on the dataset used for the ablation study of MV-VDM, and the results are reported in the table below.
>
> | Method |**I2V $\uparrow$** |**M. Sm. $\uparrow$** | **Dy. Deg.** | **Aest. Q. $\uparrow$** |
> | --- | ---| ---| ---| ---|
> | SVD | 0.927 | 0.985 | 0.849 | **0.538** |
> | Ours | **0.935** | **0.988** | 0.710 | 0.532 |
>
> **Q5: Issue about the basketball interaction in Fig. 4**
>
> Yes, it has the interaction. Please refer to the Figure 3 in our attached PDF.

---

> > ### Comment · Reviewer_Tnsn · 2024-08-14
> >
> > Thanks for your response. It has addressed my concerns. I will keep my score.

---

### Author Rebuttal · Authors · 2024-08-07

We thank for the reviewers' appreciation of our work, as they give positive comments of "problem setting is interesting, well-motivated and straightforward" (R2, R4), "achieve state-of-the-art performance of 4D generation" (R3, R5), "the large-scale 4D dataset could have significant influence on this area" (R1, R2, R3, R4, R5), "paper is well-written" (R1, R3, R5). Below we respond to some common concerns raised by the reviewers.

**Many reviewers ask for more visualizations of our methods beyond the demo video we provided. Due to the anonymity policy, we cannot share the link to our project page with more than 100 $1024\times1024$ resolution animation videos, so we provide some high-quality images in the attached PDF**. Additionally, we validate **the Gaussian trajectory learned by our model is quite accurate** and even could be used to **directly animate the MESH**, obtaining **animated mesh** that could be used in standard 3D rendering pipelines. (See Figure 6 in the PDF attached. Text prompts there are "A dragon head is roaring", "A cute dog is walking" and "A cute rabbit is dancing" from left to right.)

**1. Comparison methods**: Since we propose **a new task** of animating any off-the-shelf 3D object and there are **no previous methods** specially designed for this task, we compared our work with two state-of-the art 4D generation methods of different categories (4Dfy (4DGS) and DG4D). Although we think our experiments in the paper are solid enough to support our claims, we further provide the additional comparisons with all the methods requested by the reviewers except for AYG, which is not open-released and hard to be reproduced during the limited rebuttal period. We use official implementations of all comparison methods, and load our pre-trained static NeRF/3D Gaussians. Note that NeRF-based methods usually require training time from hours to several dozens of hours. Additionally, we add **CLIP-I** as the evaluation metric and provide results of **16-frame version of our model** as suggested.

**Quantitative results are as below and qualitative results are depicted in Figure 8 in PDF attached.** (We indicate the best and second best results in bold and italics.)

|       | **I2V $\uparrow$** | **M. Sm. $\uparrow$** | **Dy. Deg.** | **Aest. Q. $\uparrow$** | **CLIP-I $\uparrow$** |
|-------|-------------------|----------------------|--------------|------------------------|------------|
| 4Dfy (4DGS) | 0.783             | **0.996**       | 0.0          | 0.497                   | 0.786      |
| 4Dfy (NeRF)  | 0.817             | 0.990                 | 0.010        | 0.549                   | 0.834      |
| Animate124   | 0.845             | 0.986                 | 0.313        | 0.563                   | 0.845      |
| 4DGen        | 0.833             | *0.994*    | 0.187        | 0.453                   | 0.776      |
| TC4D         | 0.856             | 0.992                 | **0.830** | 0.565                   | 0.859      |
| Dream-in-4D  | 0.938             | *0.994*    | 0.0          | 0.551                   | 0.895      |
| DG4D         | 0.898             | 0.986                 | 0.477        | 0.529                   | 0.860      |
| Ours (8-frame)| *0.982*| 0.991                 | 0.597        | **0.581**         | **0.946**|
| Ours (16-frame)| **0.983**  | 0.991                 | *0.750*| *0.572*      | *0.937*|

As the comparison methods are **not specifically designed for this task**, i.e., they do not take multi-view attributes of the given 3D object into consideration, they perform not well in persevering the identity of the object (indicated by **I2V** and **CLIP-I**). Besides, they also struggle with learning motion with relatively large amplitude since they have to balance between appealing appearance and large motion (indicated by **Dy. Deg.**). **TC4D is a special case as it takes pre-defined object trajectory as the global motion**.

Our methods are superior to comparison ones in terms of both appearance and motion. **Specially, we find our 16-frame version model can generate motion with larger amplitude while having similar appearance preservation ability as our 8-frame version model. Please refer to PDF attached for a better understanding.**

**2. Ablation of MV2V-Adapter**: We ablate the MV2V-Adapter as follows. MV2V-Adapter improves almost all metrics. The decrease in **Dy. Deg.** is because w/o MV2V-Adapter the generated results are noisy and the motion is incoherent.  Qualitative comparison is in Figure 5 in PDF attached (Text prompt: "A flame rock monster is launching an attack".)


| Method | **I2V $\uparrow$** | **M. Sm. $\uparrow$** | **Dy. Deg.** | **Aest. Q. $\uparrow$** |
|---|---|---|---|---|
| w/o MV2V-adapter | 0.927 | 0.986 | 0.961 | 0.526 |
| Ours | **0.935** | **0.988** | 0.710 | **0.532** |


**3.  Motion Diversity and Dataset**: We present more generated results in **Figure 7 in PDF attached**, demonstrating that the model trained on our dataset can perform various diverse animations on the same 3D objects. This proves the generalization capability of our dataset. Admittedly, our MV-Video dataset contains only 38K animated objects with 84K animations, which is still smaller and limited compared to 2D video datasets.

However, it is worth noting that our multi-view video (4D) dataset is more rare and challenging to obtain compared to web-scale 2D video datasets, especially with regard to the multi-view camera parameters. Our dataset is much larger than previous 4D datasets. Large-scale 4D dataset is crucial for learning multi-view spatiotemporal consistent 4D foundation model, which cannot be achieved with 2D video datasets.

We've created a 4D dataset and confirmed its effectiveness. We are expanding the dataset and plan to release it, and believe it will grow  with community support, driving advancements in the 4D domain. We hope the reviewers will consider this.

---

### Decision · Program_Chairs · 2024-09-25

**Decision:**

Accept (poster)

**Comment:**

Pre-rebuttal, reviewers praised the overall relevance and novelty of the tackled problem, and the intuitive solution proposed to solve it. Recurring concerns included insufficient evaluation against too few baselines, missing ablations, concerns about the diversity of the generated motions, and incomprehensibility of the method section.

Authors provided a strong rebuttal. All reviewers unanimously agreed to accept the paper conditioned on:
1. Authors include additional comparisons and ablations in their final version.
2. Significantly improve the writing quality in the method section.

The AC thus accepts the paper conditioned on the fact that authors fix both 1. and 2. above in the final version of their paper.